



# What drives landslide risk: Disaggregating risk analyses, an example from the Franz Josef and Fox Glacier Valleys, New Zealand

Saskia de Vilder[1], Chris Massey[1], Biljana Lukovic[1], Tony Taig[2], Regine Morgenstern[1]

[1]GNS Science, Lower Hutt, 6012, New Zealand
[2]TTAC Ltd, Cheshire, CW9 6EU, United Kingdom

*Correspondence to*: Saskia de Vilder (s.devilder@gns.cri.nz)

**Abstract.** We present a quantitative risk analysis (QRA) case-study from the Franz Josef and Fox Glacier Valleys, on the West Coast of the South Island, New Zealand. The Glacier Valleys are important tourist destinations that are subject to landslide hazards. Both valleys contain actively retreating glaciers, experience high rainfall, and are proximal to the Alpine Fault, which
is a major source of seismic hazard on the West Coast. We considered the life safety risk from rockfalls, soil/rock avalanches and flows that are either seismically triggered or occur aseismically. To determine the range in risk values, and dominant contributing variables on the risk, we modelled nine different risk scenarios where we incrementally changed the variables used in the risk model to account for the underlying uncertainty. The scenarios represent our central estimate of the risk, e.g. neither optimistic nor conservative, through to our upper estimate of the risk. We include in these estimates the impact of time-
variable factors, such as a recently reactivated landslide has had on locally increasing risk and the time-elapsed since the last major earthquake on the nearby Alpine Fault. We disaggregated our risk results to determine the dominant drivers in landslide risk, which highlighted importance of considering dynamic time variable risk scenarios and the changing contributions to risk from aseismic versus seismic landslides. A detailed understanding of the drivers of landslide risk in each valley is important to determine the most efficient and appropriate risk management decisions.

## 1 Introduction

High mountain areas are subject to a variety of natural hazards, including slope instability. Globally, these mountainous areas are currently experiencing declining low elevation snow cover, retreating glaciers and degrading permafrost as a result of climate change (cf. Hock et al., 2020). Such changes in environmental, meteorological and geomorphological conditions may
influence the rate, size and characteristics of landslide hazards (Gariano and Guzzetti, 2016). Additionally, such high mountain areas are subject to seismic hazards, including seismically triggered landslides. Given that the exposure of people and infrastructure to landslide hazards is also increasing from population growth, tourism, and socio-economic development (Hock et al., 2020), the risk from landslides may change and increase with time.

Quantitative risk analysis (QRA) is an important tool for assessing, managing and communicating the risks from landslide
hazards (Corominas et al., 2014; 2015), and there is an increasing need to undertake QRA from legislative authorities and from





within the engineering and engineering geological communities (Corominas et al., 2014; Van Westen and Soeters, 2006; Ho et al., 2000). However, the ability to estimate quantified levels of risk is often challenging as the input datasets used in risk analyses are inherently uncertain. Such uncertainty is mainly due to the lack of completeness, quality, or range within the input datasets required to undertake a QRA (Van Westen and Soeters, 2006). A landslide inventory, which details where landslides

have occurred in the past, provides information critical to understand what triggers landslides, what makes a particular slope more susceptible to landsliding, and how frequently landslides are likely to occur (Guzzetti et al., 2012). Yet, for many landslide prone areas this spatial and temporal record of landsliding is limited or does not exist. This is particularly the case for certain trigger events, such as earthquakes, where the return period of the trigger event may be greater than the length of the historical record (van Westen et al., 2008). Consequently, assessments of landslide susceptibility and frequency rely heavily

on practitioner experience and judgement (Lee, 2009), and may not always reveal the full levels of uncertainty attached to the risk estimates (Corominas et al., 2014; Macciotta et al., 2015). Most approaches use past landslide behaviour to predict what may occur in the future based on the maxim "the past is key to the future" (Varnes, 1978). However, the present or future conditions that make a slope susceptible to, or trigger landsliding may be different to those of the past. Changes in the location and the frequency of landslide activity may substantially alter the estimated risk, adding to the uncertainty associated with the

risk value.

Fell et al., (2005) suggest using sensitivity factor analysis as a tool to understand the influence of potential uncertainties on the estimated risk levels, and communicate the influence of this input variability to users of the risk analysis and assessment. A current limitation of risk analysis is the need to be able to 'disaggregate' the risk results in order to determine the importance

of the different input factors included in the QRA, such as the annual frequency of a given landslide type and volume occurring under a given set of triggers, how far landslide debris travels down a slope, where people are present on the slope and their biophysical vulnerability if present and hit by landslide debris. Such limitation means that the contribution to the risk and sensitivity of the results relating to the input variables used are rarely quantified, thus making it difficult for risk managers to understand and implement targeted risk reduction measures and risk communication options. We address this here, by

presenting the QRA results, and their uncertainties, from a local to regional scale (1: 10,000 – 1: 50,000) analyses of landslide hazards and the risk they pose to the lives of people visiting and working in the Franz Josef and Fox Glacier Valleys, located on the West Coast of New Zealand.

## 2 Study Site

The Glacier Valleys, which are important tourist destinations, are located on the West Coast of the South Island, New Zealand (Figure 1). Both valleys contain multiple trails (walking and/or cycling tracks), which take between 30 minutes and up to 8 hours to walk/cycle, that allow visitors to easily access and experience a glacier environment. Within this environment, visitors





are exposed to a variety of landslide hazards. Numerous near-misses have been documented, and two fatalities occurred in January 1980 when a debris avalanche occurred along a track in the Fox Glacier Valley. Currently, the northern road and access track within Fox Glacier Valley is closed due to repeated damage from debris flow events. Evidence of landsliding is present within each valley, with the types of landslide broadly classified into rock falls, slides and topples, debris and rock avalanches and debris flows (as classified by Hungr et al., 2014). In addition to these broad landslide types, deep-seated gravitational slope deformations (DSGSD's) can be observed in both study areas. These large DSGSD's typically provide sources of material for smaller rockfall/debris avalanches or debris flows (Cody et al., 2020). Earthquakes are potential triggering mechanisms for landslides, as both study areas are located less than 10 km southeast of the Alpine fault (Figure 1), which is a major source of earthquakes in New Zealand. Additionally, both valleys experience high rainfall, with 5 m/year recorded in Franz Josef village increasing to >10 m/year towards the main divide of the Southern Alps (Langridge et al., 2016). The glaciers in each valley are currently retreating (Purdie et al., 2015,2021), exposing more disturbed and consequently weaker rock masses, which appear to be the source of many recently documented landslides. Many of these 'aseismic' landslides appear to be triggered by intense rainfall, however, several have no documented trigger. Therefore, the slopes in the study areas have and will continue to be subjected to transient changes in stress, typically caused by precipitation-induced variations in pore-water pressure, erosion, freeze-thaw cycles, and diurnal and seasonal temperature variations. Transient stress changes within a slope can lead to deformation, fracturing, and joint dilation, thus reducing rock mass strength leading to failure.

The study areas (shown in Figure 1) are dominated by ice-free slopes comprised of schist (Cox and Barrell, 2007). The structural geology of the bedrock schist is complex, given the proximity of both sites to the Alpine Fault (Figure 1). Large persistent faults cut through the area trending north-east to south-west and east to west. The quality of the rock mass is highly variable over the study areas and tends to change with proximity and location relative to these persistent faults. Moraine and colluvium deposits are present within the main and tributary valleys, with the valley floors formed of predominantly alluvium, and re-worked moraine and colluvium. The glaciers have carved the valleys, resulting in steep bedrock valley sides truncated by deeply incised streams. Debris fans are present at the mouth of these incised streams, which feed into the Fox and Waiho (Franz Josef) rivers, respectively. Flow rates within these rivers are highly seasonal and their courses, within their respective wider valleys, change frequently.

In Fox Glacier Valley, there are more extensive and thicker debris deposits (both moraine and colluvium) and larger debris fan deposits (e.g., Yellow Creek Fan; Gomez and Purdie, 2018), indicating that debris flows and avalanches may be more prevalent. With glacier retreat, these debris deposits are free to begin creeping, and debris is available for remobilisation via debris flows (Cody et al., 2020). In contrast, Franz Josef contains less debris and is more dominated by bedrock slopes, which may have been the result of limited debris accumulation through time or the ability of erosional processes to keep pace and remove material from the valley.


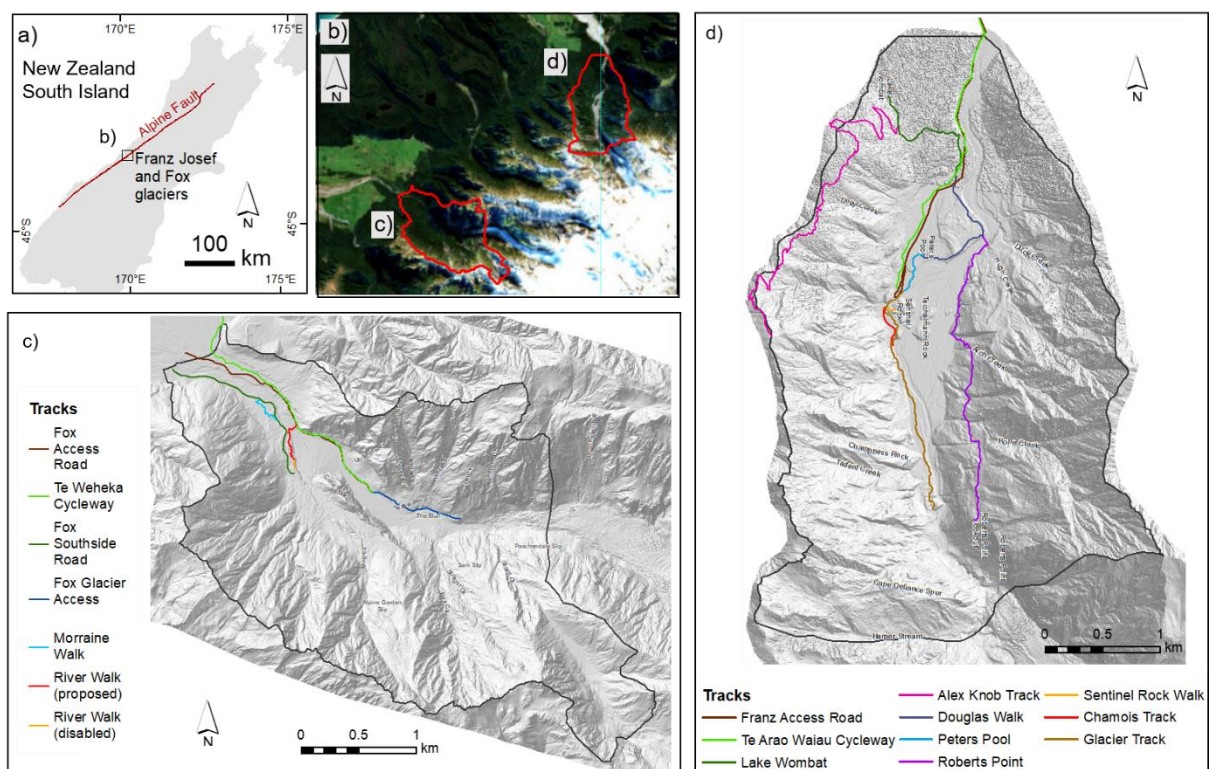

**Figure 1: Location of the Franz Josef and Fox Glacier Valleys on the West Coast of the South Island (a & b), which contains data**
**sourced from the LINZ Data service and liscensed for reuse under the CC BY 4.0 licence. c) Fox Glacier Valley, including the access**
**roads, cycleway and tracks within it. d) Franz Josef Glacier valley, including the access roads, cycleway and tracks within it.**

## 3 Methodology

### 3.1 Risk calculation route

To estimate the risk, we follow the quantified risk analyses method described in the Australian Geomechanics Society (2007),
and the Joint Technical Committee on Natural Slopes and Landslides (JTC1, as outlined in Fell et al., 2008). We calculated
the probability of death (life risk) of an individual, $P_{(LOL)}$, from:

$$P_{(LOL)} = P_{(L)} \times P_{(T:L)} \times P_{(S:T)} \times V_{(D:T)} \tag{1}$$

where:

-    $P_{(L)}$ is the probability (annual frequency) of the landslide occurring

        -    $P_{(T:L)}$ is the probability of the landslide (e.g., the debris from a landslide of a given type) reaching the element at risk

             (e.g., visitor on a track)





- P$_{(S:T)}$ is the spatio-temporal probability of the person at risk being present and in the path of landslides (the proportion of a year that the person is exposed to landslides)

- V$_{(D:T)}$ is the vulnerability of the person if present and in the path of landslide debris (i.e. the probability that the person will be killed if impacted by the landslide). In our analyses, we include in the vulnerability estimates, the potential for a person to be aware of the hazard and take evasive action.

Within the Fox and Franz Josef Glacier Valleys, we sum risk from several landslide hazards where people are exposed to: 1) Several types of landslides, 2) Landslides of the same type but different volume, 3) Landslides triggered by more than one
phenomenon, and 4) Several slopes on which landslides can occur. To take such cases into account, we re-wrote Equation 1 as:

$$P_{(LOL)} = \sum_{i=1}^{n}(P_{i(L)} \times P_{i(T:L)} \times P_{i(S:T)} \times V_{i\,D:T}) \qquad (2)$$

where n is the number of landslide hazards of a given type and volume. This assumes that the hazards are independent of each
other. However, in the valleys, it is possible that one or more of the hazards may result from the same causative event, e.g., an earthquake. Therefore, we estimate the probabilities using the theory of uni-modal bounds wherethe upper bound conditional probability (P$_{UB}$) is calculated from:

$$P_{UB} = 1 - (1 - P_1) \times (1 - P_2) \dots\dots(1 - P_n) \qquad (3)$$

where: P$_1$ to P$_n$ are the estimate of several individual hazard conditional probabilities. We then multiplied the P$_{UB}$ by the annual probability of the common causative event, e.g., the given level of shaking representing a given earthquake. More detail on the equation route is provided in the Appendices.

**3.2 Risk metrics**

We estimated the risk to life, using four risk metrics. Firstly, we estimated the local personal risk (LPR), which represents the
annual probability of death for a hypothetical person present at a particular location for 100% of the time (24-hours a day and 365 days of the year). LPR, a metric used in flooding and seismic hazard studies (Crowley, 2017; Jonkman et al., 2003, van Elk et al., 2017), can be used to visualise the spatial distribution of risk within the study areas in order to help plan/realign tracks and roads. Secondly, we estimated the individual risk per trip, which is expressed in terms of the fatality risk (probability of death) of an individual resulting from one return trip along one of the main access tracks or roads within the study areas.
We use this to represent the risk to visitors. We then estimated the annual individual fatality risk (AIFR), which is expressed in terms of the fatality risk experienced by an individual over one full year of, e.g. working in the valleys. We use AIFR to



estimate the risk to the most exposed worker in each valley who is present every day for substantial periods of the year. We estimated societal risk by determining fN pairs, which represents the frequency (f) of an accident killing (N) or more people in a single event, plotted on a fN curve (Strouth and Mcdougall, 2021). In this paper, we focus on and report the results for

LPR and individual risk per trip. AIFR and societal risk results are reported in Massey et al. (2018c).

### 3.3 Methodology framework

Our risk analysis firstly considers the possible range of triggering events in terms of a set (bands) of earthquake triggers and aseismic triggers (e.g., rain, time). In our compilation of the landslide inventories for each valley, we were unable to determine a relationship between rainfall or snowmelt with landslide occurrence. The recorded near misses in Franz Josef Glacier Valley

and two fatalities (January 1980) in Fox Glacier Valley from a debris avalanche occurred in the absence of any discernible trigger. Therefore, we subsume potential rainfall triggering, snowmelt triggering and climatic factors into an aseismic annualised rate of landsliding. However, due to the proximity of the Alpine Fault and the seismic history of the region, we explicitly considered the possibility of seismically triggered landslides.

For each representative earthquake event, we determined the annual frequency of the event and the number of landslides of a

given volume class produced in that event. For aseismic landslides, we determined the annual frequency of landslides of a given volume occurring in each valley using historical data on aseismic landslides in the valleys and the wider Southern Alps (Massey et al., 2018c). For both seismic and aseismic landslides we considered the full range of volume classes that could occur in each valley, which are: 1) $\leq 10$ m³, 2) 10 m³ to 100 m³, 3) 100 m³ to 1,000 m³, 4) 1,000 m³ to 10,000 m³, 5) 10,000 m³ to 50,000 m³, 6) 50,000 m³ to 100,000 m³, 7) 100,000 m³ to 500,000 m³, 8) 500,000 m³ to 1,000,000 m³, 9) 1,000,000 m³ to

5,000,000 m³, and 10) >5,000,000 m³. We estimated the number of landslides that could occur for each volume class using the Moon et al. (2005) method, by calculating the area under the landslide volume frequency curve (see Figure 2) using log-log histogram bins.

Secondly, we considered the locations from which landslides are most likely to source in each glacier valley. We explicitly

determined landslide source locations, in order to estimate how far the debris could travel downslope from a particular source. We used slope angles, volume to area scaling relationships and geomorphic mapping to delineate these source areas. We compiled information on pre-disposing factors of slope instability in each valley to understand spatial controls on landslide occurrence, with these datasets forming an important input into landslide susceptibility modelling for each valley (Reichenbach et al., 2018). We used logistic regression susceptibility models for both seismic (Massey et al., 2021) and aseismic landslides

to weight which source areas may preferentially generate landslides.

We conducted 3D numerical runout simulations to determine $P_{(T:L)}$: the probability of the debris from a landslide reaching or passing a portion of slope as it travels downhill from the source area. We conducted these numerical simulations for rockfall,



debris avalanches and debris flows from our explicitly determined source areas. We used RAMMS rockfall software (2015)
for rockfall simulations, and RAMMS debris flow software (2011) for debris flow and debris avalanche simulations.

We compiled information on the length of time visitors and workers spend along the tracks in each valley to estimate the
spatio-temporal probability of the person at risk being present at a location ($P_{(T:S)}$) and consequently in the path of debris. We
used empirical estimates of vulnerability (*V*), which is the probability of a person being killed if present and in the path of one
or more boulders, considering both: a) the likelihood of being killed if struck; and b) the possibility of being able to take
evasive action and avoid being struck.

For each of these steps and elements of the risk equation, we determined central estimates (which we define as neither
optimistic nor conservative and is based on taking the mean estimate) and upper estimates (based on the 84[th] percentile) of the
different variables used in the QRA. We used these different estimates in our sensitivity analysis, where we incrementally
changed the variable estimates from central to upper, until all variables were upper estimates. This results in 9 different risk
models, from which we can calculate the incremental change in risk based on varying the input assumptions and document the
impact of aleatory uncertainty.

### 3.4 The probability(annual frequency) of the landslide occurring: $P_{(L)}$

### 3.4.1 Seismic Landslides

We determined the frequency and volume of landslides likely to be generated at different magnitudes of ground shaking
intensity from the mapped landslide distributions of historical New Zealand and international earthquakes, as detailed in de
Vilder et al. (2020). We used the landslides generated during the 2016 $M_W$ 7.8 Kaikoura, 1968 $M_W$ 7.1 Inangahua and 1929
$M_W$ 7.8 Murchison earthquakes, as proxies (Massey et al., 2018b; Hancox et al., 2014, 2015). We selected these three landslide
inventories as they represent the most complete New Zealand inventories for seismic landslides that occurred in fractured hard
rock (such as greywacke) similar to that of schist, and occurred in mountainous and hilly terrain.

We assessed the number of landslides that could be generated for four different representative earthquake events, as represented
by peak ground acceleration (PGA) bands: Band 1 (0.2 – 0.35 g); Band 2 (0.35 – 0.65 g); Band 3 (0.65 – 1.2 g) and Band 4 ($\geq$
1.2 g). It is unlikely that several landslides will be generated by ground shaking < 0.2 g (Dowrick et al., 2008). We calculated
the annual frequency of the representative PGA per band from the New Zealand National Seismic Hazard Model (NSHM)
(Stirling et al., 2012), by subtracting the annual frequencies that represent the PGA boundaries (start and end) of each band.
These frequencies are time independent and do not consider time elapsed since the last earthquake on the Alpine Fault, which
occurred in 1717. Recent research (Howarth et al., 2021) shows that the probability of an earthquake occurring on the central
section of the Alpine Fault is 75% per cent in the next 50 years, and that there is an 82% chance that the earthquake will be





greater than M8. To account for this, our upper estimate of the PGA annual frequencies, we increase the annual frequency of the most intense ground shaking (Band 4) to 0.015 to reflect time elapsed since the last Alpine Fault earthquake.

To assess the magnitude-frequency of seismic landslides in each band, as outlined in de Vilder et al. (2020), we firstly
determined the appropriate landslide source volume to area scaling relationship (from Massey et al., 2020). Secondly, we estimated the landslide frequency (number) and source area scaling relationship. Thirdly, we investigated the relationship between landslide occurrence and PGA, slope angle and material type using the Kaikoura, Inangahua and Murchison landslide inventories. Finally, we combined estimates of the annual frequency of the representative event PGA for each earthquake band in the NSHM. Using this relationship, we estimated the probability of a landslide of a given volume class occurring within
each study area for each PGA band considered, along with the annual frequency of the representative PGA in the band occurring (see Figure 2). We fitted power laws to the data, with these representing our central estimate of the number of landslides of given volume class occurring for each PGA band considered. To derive an upper estimate, we added the standard error of the gradient of our best-fit power-law to the power-law relationship to calculate the number of landslides that could be generated.

**3.4.2 Aseismic Landslides**

We collated information on the occurrence of aseismic landslides from various data sources, which we used to assess the historical type, mechanisms and rates of aseismic landslides for both valleys.  These data sources include 1) a rockfall register compiled from observations made by staff of the Department of Conservation (DOC), Franz Josef Glacier Guides Ltd, and Fox Glacier Guides Ltd, 2) a landslide inventory derived from historical aerial imagery analysis of both valleys, and 3) a large
landslide inventory of historical landslides observed in the wider southern alps (see the appendices for more information on the compilation of the landslide inventories).

We determined valley-specific magnitude-frequency relationships of landslides, given the amount of catchment-specific information about landslides. We fitted power law trends to the data to generate a central estimate and an upper estimate (using
the standard error of the power law) of the number of landslides that could occur in each valley per km² and their annual frequency (see Figure 2 b). These landslide rates were then scaled to each valley by multiplying them with the total area of the slopes greater than 30° within each valley. We used a slope angle of 30° as a cut-off as we assumed the landslide types considered here, were unlikely to occur on slope angles <30°.




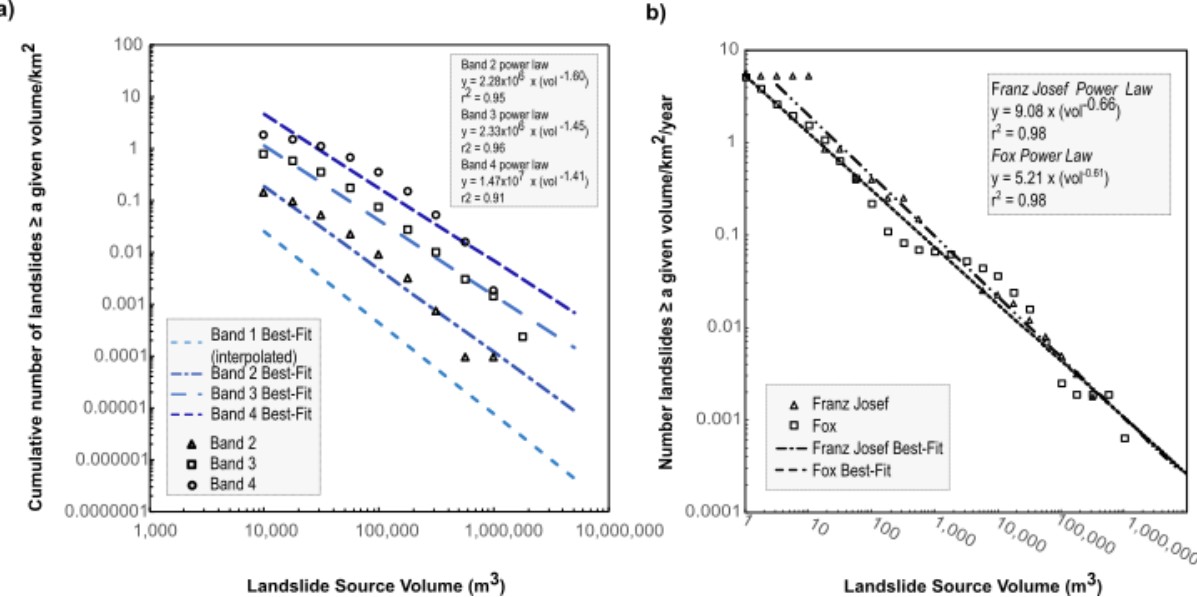

Figure 2: a) The number of landslides of given landslide volume that could be generated for different levels of ground-shaking as represented by Bands 1 through to 4. b) Magnitude Frequency relationships for aseismic landslides for Fox Glacier Valley and Franz Josef Glacier Valley. The two power laws on each graph represent the central estimate (using the power-law relationships)

## 3.5 The probability of the landslide reaching the track or road

### 3.5.1 Landslide Susceptibility

In the absence of historical information on landslides triggered by an Alpine Fault earthquake, we used the 2016 Mw 7.8 Kaikoura Earthquake-induced landslide inventory to understand the spatial controls on susceptibility to failure. From this dataset, Massey et al. (2018a, 2021) used a logistic regression model to correlate the three aforementioned mapped earthquake induced landslide inventories with various topographic, geological and seismological parameters to understand which parameters best explained the occurrence of coseismic landslides. We applied the Massey et al. (2018a) logistic regression model to both valleys, using PGA input from the NSHM.

We developed valley-specific logistic regression models to determine aseismic landslide susceptibility, given the amount of landslide-specific information and slight differences in the landslide hazards within each valley (see Figure 3 for an example from Fox Glacier Valley). These models are based on the correlation of mapped landslides, and various topographic, geological and land use characteristics (cf Massey et al., 2018a and the appendices ). Rockfalls recorded in the rockfall register were not included within the analysis as the data do not have accurate geographic locations. More information on the aseismic susceptibility models is provided in the appendices.

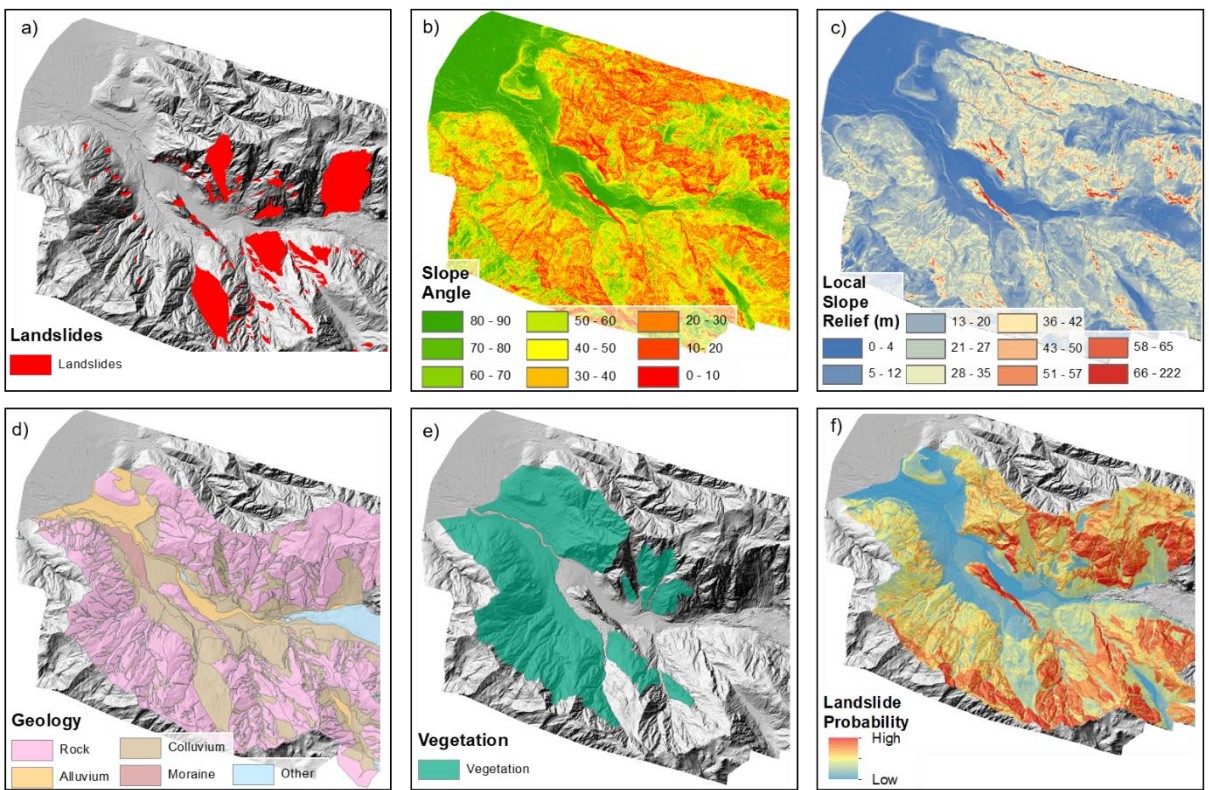

**Figure 3: Generation of logistic regression model for Fox Glacier Valley from the correlation between a) The landslide inventory, and the static variables of b) slope angle, c) local slope relief (LSR), d) geology, and e) vegetation. The logistic regression model calculates f) landslide susceptibility to failure.**

### 3.5.2 Landslide Runout

Landslide sources ≤1,000 m$^3$ were assumed to be rockfalls rather than debris flows and debris avalanches. Potential rockfall source areas were defined using all slopes ≥45º, assuming any slope ≥45º can potentially generate rockfalls (Figure 4). For landslide volumes ≤100,000 m$^3$ the landslide sources were assumed to be pixels of a given area based on the area to volume scaling exponents (Figure 4). For landslide volumes >100,000 m$^3$, the shapes of the sources were defined using the geomorphic features (Figure 4).

For the rockfall simulations, we used RAMMS rockfall software (2015), which simulates the rigid body motion of falling rocks and predicts rock trajectories in general three-dimensional terrain (see the appendices for more information). For the debris avalanches and debris flow simulations, we used RAMMS debris flow software (2011), changing the Voellmy friction parameters (see the appendices for more information) to determine if a particular source area failed as a debris avalanche or



debris flow. From these simulations, we derived the runout extent and maximum debris height. The simulated maximum height
of debris passing through a given grid cell is converted into the number of boulders (our central estimate), with our field
measurements indicating that the average boulder size is 1 m³. For example, if the maximum debris height passing through a
3 m by 3 m grid cell is 1 m, then the total volume of debris passing through that grid cell is 9 $m^3$, which when converted into
N boulders, would be on average 9 boulders. Based on sensitivity analysis of the Voellmy friction parameters (see the

appendices for more information), we calculated a standard deviation-based factor of difference in debris height. We applied
this factor of difference to the simulation results to increase debris height, providing an upper estimate.

We calculated the probability of one boulder in the debris hitting an object when passing through a particular portion of the
slope, perpendicular to the debris path, using the equation:

$$\mathbf{P_{1(T:L)}} = \frac{D+d}{L} \tag{4}$$

where D is the diameter of the boulder, d is the diameter of a person (our central estimate assumes a person is a "cylinder"
with a 1 m diameter, while our upper assumes an estimate of 2 m diameter), and L is the unit length of slope perpendicular to
the runout path, which for this study is 3 m grid cell. Our equation includes a "buffer-zone" around the person ($D + d$) within

which the boulder travels along a path either side of $d$ and cannot miss.


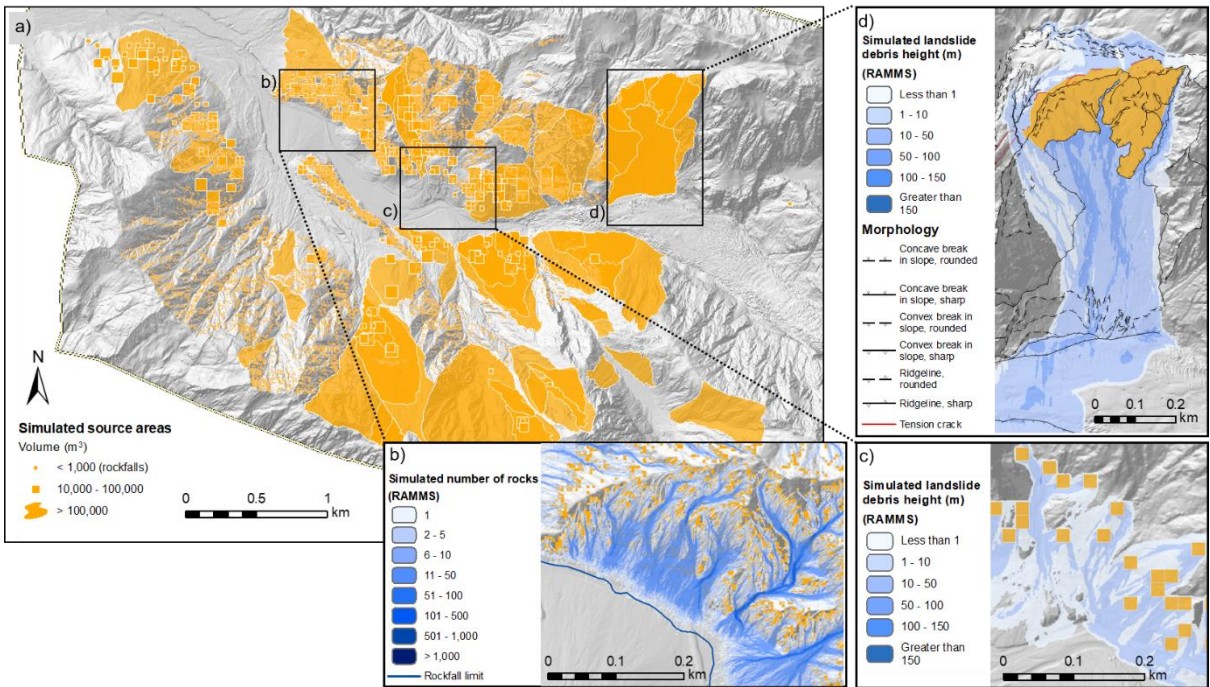

Figure 4: a) Simulated source areas in Fox Glacier Valley for all volume classes. b) Numerical rockfall simulations, using RAMMS, from pixel source areas. c) Numerical debris avalanche and debris flow simulations, using RAMMS, from pixel source areas for volume classes 10,000 m³ to 100,000 m³. d) Numerical debris avalanche simulations, using RAMMS, for geomorphically defined source areas for volume classes ≥500,000 m³.

## 3.6 Exposure

We calculated the probability that a person will be occupying a given grid cell along one of the tracks/roads ($P_{(S:T)}$) if they spend a number of hours ($N_{HRS}$) per trip per year walking/driving that route using the equation:

$$P_{(T:S)} = \frac{(N_{HRS})}{(N_C)} \tag{5}$$

Where $N_C$ is the number of cells visited along the route. We compiled information on the estimated time taken to travel by vehicle along given roads to/from the car parks, and the time taken for walking a round trip from the car park to the glacier viewing points to determine the time exposed for an average walker (central estimate) or a slow walker (upper estimate). Therefore, we assumed that the time spent on each one metre section of track was equal to the duration (time) of travel divided by the total length of the track. However, this can be adjusted to account for longer time spent by a visitor at a viewing area, picnic spots etc. For the calculation of LPR, we assumed $P_{(T:S)}$ of 1, where a person is present 100% of the time.





### 3.7 Vulnerability

Physical vulnerability (*V*) depends on the landslide intensity, the characteristics of the elements at risk, and the impact of the landslide (Du et al., 2013). To derive our central estimate of vulnerability, we link vulnerability values to representative landslide volumes, which act as a proxy for landslide intensity (see the appendices for more information). Anecdotal evidence

from the glacier valleys suggests that evasive action reduces vulnerability. This was the case on 13 October 2011 and 16 June 2014, when boulders and fly rock from debris avalanches passed over several people on the ice. During these near misses, the guides heard the debris moving down the slope and had time to instruct their clients to take evasive action. In Table 1 our vulnerability values scale with landslide volume, were for landslide volumes ≤100,000 m³ an individual may be able to take evasive action. However, the ability to take evasive action decreases with landslide volume. For landslide volumes >100,000

315 m³ an individual is likely to be buried by debris and killed. To derive an upper estimate of vulnerability, we assume a vulnerability of 1 for all rockfall and landslide volumes.

**Table 1: Physical vulnerability values used in study**

| Representative Landslide Volume (m³) | Vulnerability | |
| --- | --- | --- |
| | Central Estimate | Upper Estimate |
| 1,000 | 0.1 | 1 |
| 10,000 | 0.5 | 1 |
| 50,000 | 0.5 | 1 |
| 100,000 | 0.9 | 1 |
| ≥500,000 | 1 | 1 |

### 3.8 Rick scenarios modelled

For each valley, we estimated the individual risk per trip for a visitor using nine different risk model scenarios (Table 2) which ranged from our central estimate of the risk to our upper estimate of the risk. Our central estimate (Scenario 1: Table 2) risk model uses the central estimate input variables and a time independent ground shaking annual frequency. For each risk model scenario we incrementally change the variable from central to upper estimates, starting with the number of landslides that

could occur in an earthquake event (Scenario 2), then the number of aseismic landslides that occur annually (Scenario 3), before increasing our estimate of debris height (Scenario 4), the diameter of a person (Scenario 5), our vulnerability estimate (Scenario 6), the time elapsed since the last Alpine Fault earthquake (Scenario 7), and lastly increase the length of time a visitor is exposed to the risk (Scenario 8). We also included a risk model scenario (Scenario 9: Table 2), where we used central estimate input variables but account for the increased probability of Alpine Fault earthquake occurring, to understand the

impact of these assumptions on the risk results.





**Table 2: Risk model scenarios and associated input variables**

| Risk Scenario | Number of landslides generated during an earthquake | Annual number of aseismic landslides | Debris Height | Diameter of a person | Vulnerability | EQ Scenario | Spatio – temporal probability |
|---|---|---|---|---|---|---|---|
| 1 | Central | Central | Central | 1 | Central estimate | Time Independent | Average walker |
| 2 | Upper | Central | Central | 1 | Central estimate | Time Independent | Average walker |
| 3 | Upper | Upper | Central | 1 | Central estimate | Time Independent | Average walker |
| 4 | Upper | Upper | Upper | 1 | Central estimate | Time Independent | Average walker |
| 5 | Upper | Upper | Upper | 2 | Central estimate | Time Independent | Average walker |
| 6 | Upper | Upper | Upper | 2 | Upper estimate | Time Independent | Average walker |
| 7 | Upper | Upper | Upper | 2 | Upper estimate | Time Dependent | Average walker |
| 8 | Upper | Upper | Upper | 2 | Upper estimate | Time Dependent | Slow Walker |
| 9 | Central | Central | Central | 1 | Central | Time Dependent | Average Walker |

## 4 Results

### 4.1 Individual risk per trip

The individual risk per trip in Franz Josef Glacier Valley ranged from $7.8 \times 10^{-7}$ (central estimate: Scenario 1) to $8.3 \times 10^{-6}$ (upper estimate: Scenario 8). The risk along the road in Franz Josef ranges from $7.88 \times 10^{-9}$ to $1.03 \times 10^{-7}$, while the risk along the track in Franz Josef ranges from $7.72 \times 10^{-7}$ to $1.08 \times 10^{-5}$. For Scenario 9 (central estimate with higher earthquake annual frequency), the risk along the road in Franz Josef was $6.84 \times 10^{-8}$ and the risk along the track was $2.73 \times 10^{-6}$, with a total risk per trip of $2.8 \times 10^{-6}$. The individual risk per trip in Fox Glacier Valley ranged from $4.9 \times 10^{-6}$ (central estimate: Scenario 1) to $1.7 \times 10^{-5}$ (upper estimate: Scenario 8). The risk along the road in Fox ranges from $2.57 \times 10^{-7}$ to $6.34 \times 10^{-7}$, while the risk along the track in Fox ranges from $4.63 \times 10^{-6}$ to $1.62 \times 10^{-5}$. For Scenario 9 (central estimate with higher earthquake annual frequency), the risk along the road in Fox was $4.22 \times 10^{-7}$ and the risk along the track was $7.16 \times 10^{-6}$, with a total risk per trip of $7.59 \times 10^{-6}$. The risk along roads is less than that along tracks; this is a function of both overall lower LPR risk, and less time spent on the roads. It is important to note that the risk numbers reported here do not consider any risk management and mitigation so should not be treated as indicative of current residual risk levels following actions taken in light of this analysis.

### 4.2 Risk disaggregation

Using Scenario 1, we disaggregate our risk results to understand the contributions to risk from the different risk model components. Figure 5 and Figure 6 display an LPR map of Franz Josef and Fox respectively, illustrating the spatial variation in risk within the valley and along the access tracks to the viewpoint of the glaciers. Aseismic landslides account for 66% and





83% of total LPR along the access tracks in Franz Josef and Fox, respectively, in contrast to 34% and 17% for seismic landslides (Figure 5 b & 6 b). In Fox, increases in aseismic landslide risk are observed when the track is close to the base of larger, steeper slopes or crosses a large debris fan (Figure 6 a). Although the risk to an individual is higher for aseismic landslides, the risk of a large landslide causing multiple fatalities or multiple landslides occurring at the same time leading to multiple fatalities, is dominated by earthquake events with PGA's > 0.6 g (Band 3 & Band 4) (cf. Massey et al., 2018c). For aseismic landslides, we disaggregated the risk further to determine which landslide volume classes contributed most to the risk. In Franz Josef, moderate sized landslide volume classes of 10,000 m³, 50,000 m³, and 100,000 m³ account for 31%, 30%, and 14% of the aseismic landslide risk along the track, while landslide volume classes of 500,000 m³ and 1 M m³ account for a further  7%, and 11% of LPR, respectively (Figure 5 c). Landslide volume classes of 5 M m³ or greater account for less than 3% of LPR. The risk from rockfalls (4% of LPR along track), increases when the track is closer to the base of the steep valley sides as displayed in the spikes in risk associated with volume classes of 10 m³ (Figure 5 c). Increases in the risk associated with 10,000 m³ landslide volume is associated with increases in both aseismic and seismic risk along the track (Figure 5). In Fox Glacier Valley, landslide volume classes of 10,000 m³, 50,000 m³, and 100,000 m³ account for 24%, 21%, and 12% of LPR, respectively (Figure 5 d).  Larger volume classes of 500,000 m³, 1 M m³, 5 M m³ and > 5 M m³ contribute 12%, 15%, 10% and 6% to LPR, respectively. For seismic landslides in Franz Josef, Band 2 contributes the most to the risk, accounting for 43% of LPR, while Band 1 accounts for 21%, Band 3 for 32% and Band 4 for 4% of LPR (Figure 5 d). A similar pattern exists for seismic landslides in Fox, where Band 2 contributes the most risk, accounting for 48% of LPR, while Band 1 accounts for 29%, Band 3 accounts for 21% and Band 4 accounts for 2% of LPR (Figure 6 d).




**Figure 5: Risk results from Scenario 1 (Central Estimate) risk model for Franz Josef Glacier Valley. a) LPR map displaying areas of higher and lower risk, along with the location of tracks (black dotted lines) in the valley. The risk results presented in b) to d) are extracted along the lower track in a). b) LPR values for aseismic landslides compared with seismic landslides, c) LPR values for different volume classes of aseismic landslides. d) LPR values for the different bands of ground shaking (from lowest Band 1 through**
**to highest Band 4).**



Figure 6: Risk results from Scenario 1 (Central Estimate) risk model for Fox Glacier Valley. a) LPR map displaying areas of higher and lower risk, along with the location of the access track (black dotted line) in the valley. The risk results presented in b) to d) are extracted along the track in a). b) LPR values for aseismic landslides compared with seismic landslides, c) LPR


values for different volume classes of aseismic landslides. d) LPR values for the different bands of ground shaking (from lowest Band 1 through to highest Band 4).

However, in Scenario 9 (Table 2), we modelled the increased annual frequency of a large Alpine Fault event that was assumed to result in the greatest ground shaking (Band 4), while using central estimate for all other input variables to the risk model. In this scenario, the seismic landslide risk in Franz Josef (Figure 7 a) is higher, accounting for 81% of LPR along the track, than

that of aseismic landslide risk in contrast to the patterns in Scenario 1 (Figure 5 b). In Fox, the contribution of seismic landslide risk is higher in Scenario 9, accounting for 46% of LPR along the track (Figure 7 b), than in Scenario 1 (Figure 6 b), and in locations along the track, surpasses that of aseismic landslides. However, aseismic landslides contribute more to the overall LPR (54%), particularly in locations where the track crosses debris fans (Figure 7 b).

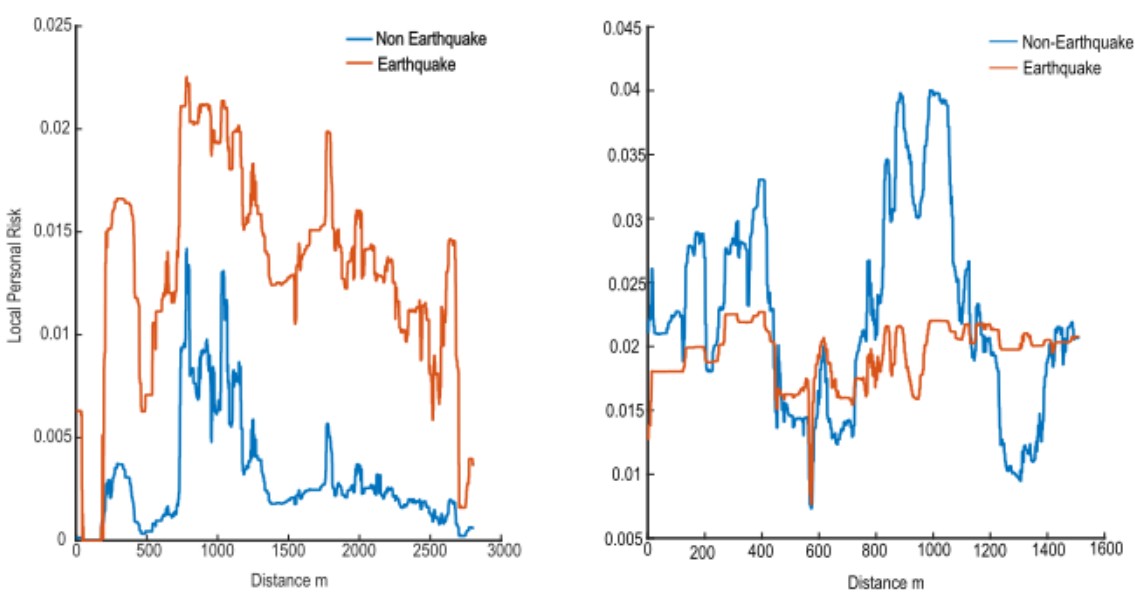


**Figure 7: Risk results for Scenario 9 (central estimate & time dependent earthquake frequency) along the access track in a) Franz Josef Glacier Valley and b) Fox Glacier Valley, displaying the contributions of aseismic landslide risk and seismic landslide risk.**

For the eight different risk scenarios (Table 3), we calculate the overall cumulative increase in risk as a percentage, along with

the amount of cumulative increase in risk between each scenario. We also calculate the increase in risk between each scenario as a percentage, to understand the contribution of each variable to overall risk in order to negate the effect of the order in which each variable is altered in the risk models and the compounding effect of changes to variables on the cumulative risk results. Our sensitivity analysis of the risk model inputs for Franz Josef (Table 3) shows that, within the ranges of inputs considered, the largest increase in the risk is associated with increasing the number of aseismic landslides that occur annually with a

cumulative increase in risk of 365%. Second to this, is the increased earthquake annual frequency, with a cumulative increase



in risk of 330%. Thirdly, increased exposure time results in a cumulative risk increase of 328% while a constant vulnerability of 1 results in a cumulative increase in risk of 219%. Changes in input variables (debris height and diameter of a person) that affect the $P_{(T:L)}$ term resulted in negligible changes to risk, with < 10% change in cumulative risk. Overall, changes in the input variables from central estimate to upper estimate resulted in a 1298% cumulative increase (just over an order of magnitude) in

the risk results. For Fox, increases in exposure time and vulnerability resulted in the largest increase in risk ( increase in cumulative risk of 80% and 60%, respectively: Table 3), while changes in the annual frequency of earthquake events as well as the number of aseismic landslides resulted in cumulative increases of risk of 56% and 30%, respectively (Table 3). Changes in the number of seismic landslides resulted in a cumulative increase in risk of 10%. Similarly, to Franz Josef, changes in debris height and the diameter of a person had negligible impact (Table 3). The range in risk values for Fox from central

estimate to upper estimate was smaller than for Franz Josef, with a cumulative percentage of increase of 244%.

**Table 3: Risk Model Sensitivity Analysis displaying the factor in increase risk between each scenario and the cumulative increase in risk from central to upper estimate (Scenario 8).**

| Risk Model Scenario | Changing risk variable | Franz Josef Glacier Valley | | Fox Glacier Valley | |
|---|---|---|---|---|---|
| | | Cumulative Risk Increase Percentage % | Cumulative Risk Increase Percentage % between scenarios | Cumulative Risk Increase Percentage % | Cumulative Risk Increase Percentage % between scenarios |
| 1 | Central Estimate Scenario | NA | NA | NA | NA |
| 2 | Increased number of landslides generated during an earthquake | 46 | 46 | 10 | 10 |
| 3 | Increased number of aseismic landslides | 411 | 365 | 40 | 30 |
| 4 | Increased debris height | 420 | 9 | 42 | 2 |
| 5 | Increased diameter of a person | 421 | 1 | 42 | 0 |
| 6 | Increased vulnerability | 640 | 219 | 108 | 66 |
| 7 | Increased earthquake annual frequency | 970 | 330 | 164 | 56 |
| 8 | Increased exposure time | 1298 | 328 | 244 | 80 |

**4.3 Changing risk through time**

Recently within the Fox Glacier Valley, there has been increased debris flow events from the Mill's Creek catchment. The debris flows are sourced from the toe of the Alpine Gardens landslide, which is an approximately ~50 million m$^3$ actively moving landslide complex in the Fox Glacier Valley (Figure 8). These debris flows travel down Mill's Creek and deposit on the debris fan at its confluence with the Fox River. The debris flow activity has resulted in the expansion of the Mill's Creek

debris fan, which in turn, has forced the migration of the Fox river to the true-right side of the valley.





This is changing the rate of debris flow activity, and concentration in a specific area influences landslides susceptibility and magnitude-frequency. The change to the landslide hazard has an impact on the estimated risk levels. The increase in activity from a particular area is a common phenomenon, based on the long-term observations of national park staff and glacier guides

(Marius Bron- personal communication), with this type of behaviour described colloquially as "switching on – and off", whereby a particular gully or slope will display enhanced rates of landslide activity for a period of time (sometimes in the order of years) before the levels of activity reduce. We incorporated the elevated debris flow activity into the risk analysis, by deriving a specific magnitude-frequency relationship for debris flows from the Alpine Gardens and Mill's Creek catchment and applying this revised magnitude-frequency relationship to source areas within this catchment (see the appendices for

information).

The increased landsliding from the Alpine Gardens area was propagated through the risk equation. Figure 9 displays the LPR map that includes the current elevated rates of debris flow activity in the Fox Glacier Valley. In the Alpine Gardens – Mill's Creek catchment, the increase in LPR ranges from 5% to 1442%, with a mean increase in LPR of 285.5% ± 245%.


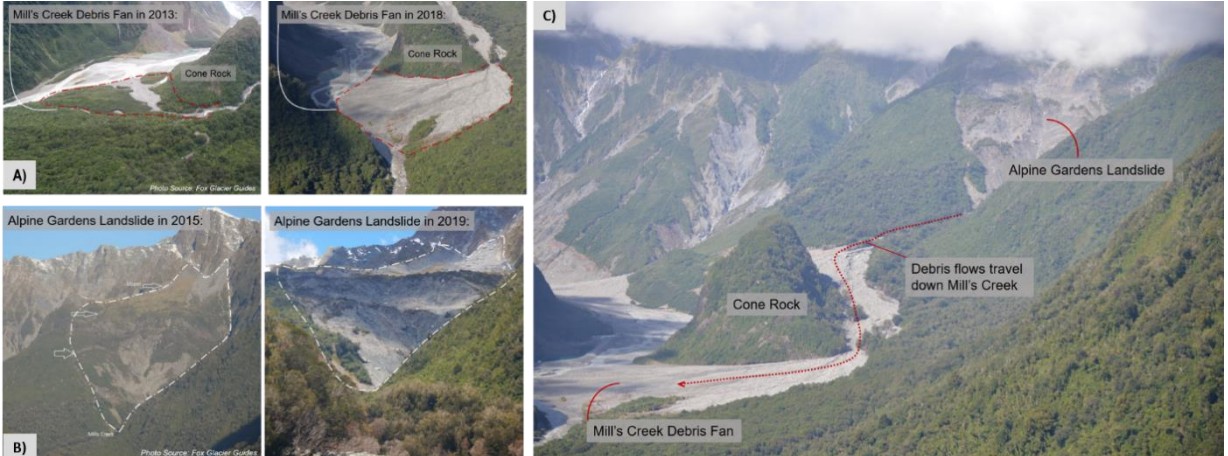

**Figure 8: a) & b) Photographs of the evolution of the Mill's Creek debris fan and Alpine Gardens landslide. c) The linkage between the Alpine Gardens landslide and Mill's Creek debris channel and fan.**


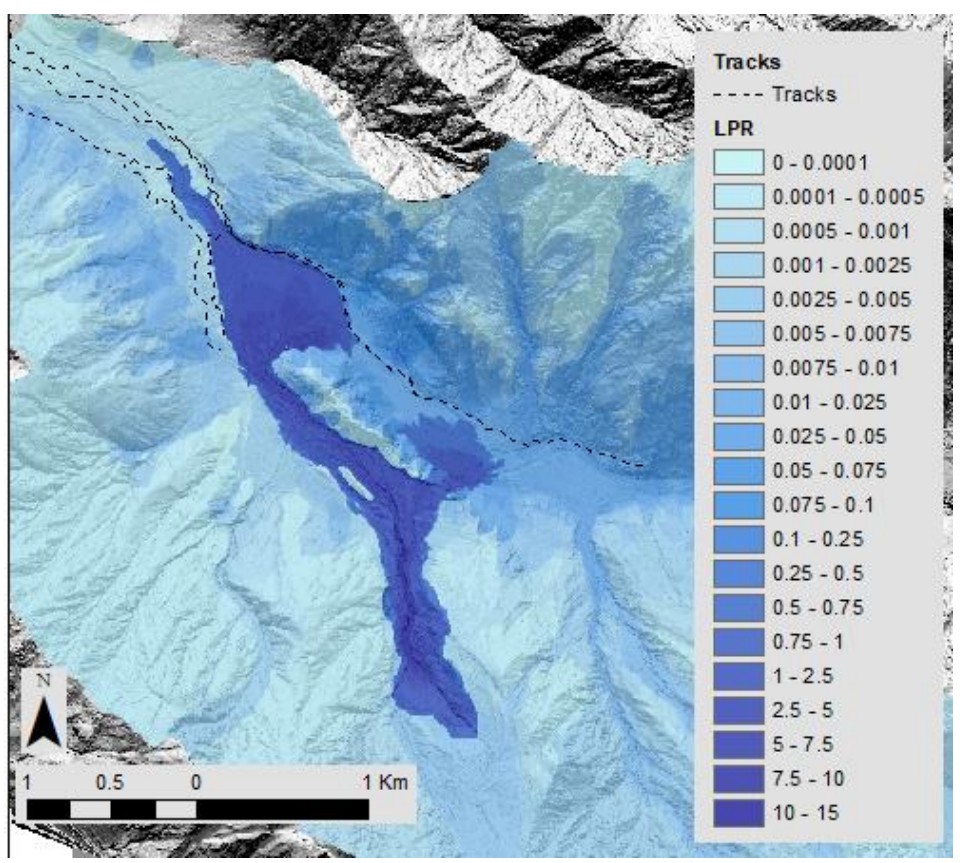


**Figure 9: Calculation of local personal risk for Fox Glacier Valley, including recent elevated levels of activity in the Alpine Gardens – Mill's Creek debris complex.**

## 5 Discussion

### 5.1 Drivers of risk

We disaggregate our QRA to determine the dominant contributors to risk in each Glacier Valley. For both valleys, in our central estimate scenario, aseismic landslides dominate the risk profile. Of these aseismic landslides, the major contributors to the risk are the moderate sized landslides (10,000 m³ to 100,000 m³), which happen more frequently than the large or very large landslides (> 100,000 m³) but travel further and impact a larger area than the more frequently occurring small landslides (<10,000 m³). Only when the tracks veer closer to the base of the slope does the risk from small landslides and rockfalls increase. We suggest that a similar pattern would be observed for seismic landslide volumes, given that the same volumes will impact the same area and that increases in the risk associated with 10,000 m³ landslide volume class in Franz Josef is associated with increases in both seismic and aseismic LPR.





Even when the annual frequency of a large ground shaking event is increased to account for the probability of > Mw 8 Alpine Fault event occurring in the next 50 years (Howarth et al., 2021), aseismic landslides account for more than half of the risk in Fox Glacier Valley. In Franz Josef Glacier Valley, increases in the number of aseismic landslides result in a large increase in risk of 365%, suggesting that within the aleatory uncertainty (e.g. within the standard deviation) of the landslide magnitude-frequency relationships, aseismic landslides are a dominant contributor to the risk. Similar conclusions were reached by

Robinson et al. (2016) in their analysis of co-seismic landsliding from an Alpine Fault event, which suggests that for the central section of the Southern Alps aseismic erosional processes are more important than seismically driven landslide erosion on annual time-scales.

However, accounting for the increased probability of an Alpine Fault earthquake occurring in the next 50 years and increasing

the number of landslides that could occur during an earthquake event each lead to increases in our risk estimates, both the individual risk per trip, discussed in detail here, and societal risk. In Franz Josef, increasing earthquake annual frequency and the number of seismic landslides resulted in increases in risk of 330% and 46%, respectively. The increase in earthquake annual frequency results in a larger cumulative increase in risk of 330% compared to 46% for the increased number of seismic landslides – the difference may be the result of the order of the scenarios and compounding effect of variables. In Fox,

increasing the annual frequency of earthquakes and the increasing number of aseismic landslides both result in risk increases of 56% and 30%, respectively. Increasing the number of seismic landslides results in risk increases of 10%, lower risk increases than that observed in Franz Josef. In Fox Glacier Valley, the presence of large debris fans indicates debris flow activity (Gomez and Purdie, 2018; e.g. Cody et al., 2020), however, debris flow records in both valleys are limited and therefore the aseismic debris flow risk may be underestimated. Our example from Mill's Creek debris fan highlights that local increases in debris

flow activity can significantly affect the risk, with local increases in risk of up to 1442%. For both seismic and aseismic landslides, the impact of the number of landslides generated, which is the $P_{(L)}$ term in the risk equation, emphasises the importance of the landslide inventory as an input into the risk calculation process. Therefore, more time and resources dedicated to the creation of a landslide inventory may reduce the uncertainty associated with the risk values (van Westen et al., 2008).


The biggest increase in risk values in Fox is associated with increases in the vulnerability and spatio-temporal probability of a visitor being in the path of a landslide (66% and 80% increases in risk respectively), with these factors resulting in increases in risk in Franz Josef (219% and 328%, respectively). This emphasises the importance of risk management decisions to reduce exposure and lower vulnerability. Changes to the diameter of a person and debris height had a very limited impact on the

estimated risk values, which affect the $P_{(T:L)}$ term in the risk equation. Changes in the spatial extent of debris from the numerical simulations were not included within the sensitivity analysis, but could be included using empirical or other probability based (e.g. Flow-R: Horton et al., 2013) runout analysis and calculations of runout probability of exceedance (McDougall, 2017; e.g.



Brideau et al., 2020). In this case-study, we assume such variations are not particularly meaningful given the number of source areas the rockfall and landslide runouts were simulated from (Figure 2), the confined nature of the valleys, and the proximity
of the access tracks and road to the base of the steep slopes from which the debris is sourced.

## 5.2 Time variable risk

Our sensitivity analysis highlights the importance of accounting for time-variable risk with the inclusion of the increased frequency of an Alpine Fault earthquake resulting in cumulative increases in risk of 330% and 56% for Franz Josef and Fox Glacier Valleys, respectively. Alongside this, increases in aseismic landsliding results in cumulative increases of risk of 365%
and 30% for Franz Josef and Fox Glacier Valley, respectively. Use of the upper estimate of the number of aseismic landslides that could occur may represent a future climate change scenario, reflecting the increased rates of landsliding (Gariano and Guzzetti, 2016) as glaciers retreat, slopes debuttress and the environmental condition changes. Our sensitivity analysis suggests that climatically driven increases in landsliding will have a larger impact on landslide risk in Franz Josef than in Fox Glacier Valley. We hypothesise that the differences in sensitivity analysis between the valleys may reflect the geomorphology of each
valley. In Franz Josef, increases in the number of larger landslides may significantly increase the probability of such landslides reaching the element at risk.

However, the size and frequency of landslides may change in response to climate change (Huggel et al., 2012; Korup et al., 2012), with Liu et al. (2021) observing a shift in the frequency-area distribution with larger landslides occurring in their dataset
of landslides in the high mountains of Asia. The larger debris and alluvial fans in Fox may indicate higher rates of aseismic landsliding than those observed in Franz Josef, and consequently may also explain that due to the already high rates of landsliding, increases in landslide rates (both seismic and aseismic) may have limited impact on risk, while the changes in vulnerability and exposure have a relatively bigger impact on the overall risk value.

Changes in landslide susceptibility should also be accounted for, as highlighted by Reichenbach et al. (2018), where the spatial pre-disposition to landsliding may change in response to environmental changes though the exact changes to landslide susceptibility are unknown. Given that landslide susceptibility is usually the starting point for risk analyses, time-variable landslide and therefore susceptibility means that the risk to people and infrastructure from landslides is also time-variable, especially after a major earthquake (e.g. Massey et al., 2014; Lin et al., 2006; Marc et al., 2015). Following the Canterbury
earthquake sequence in New Zealand, a time-varying seismic hazard model was used as input to quantify the risk to life from rockfall in the Port Hills of Christchurch (cf. Massey et al., 2014). Consequently, the rockfall risk was shown to be time-variable with a rapid 50% decrease in seismic rockfall risk in the 5 years post earthquake event and a 14% decrease in risk 5 to 10 years post event. Massey et al. (2022) shows that aseismic rockfall risk is also elevated post-earthquake event with a similar 50% decrease in rockfall rates 1 to 5 years post earthquake event. Our example of increased debris flow activity on the
Mill's Creek debris fan allows for spatial changes in landslide susceptibility frequency and magnitude to be easily incorporated





into the risk model. Our analysis shows that the impact on LPR on the Mill's creek debris fan is significant. The ability to dynamically update the risk model to account for increased landslide activity in a specific area or catchment, allows changes in environmental conditions and progressive failure, for example, increased number and size of landslides (Purdie et al., 2015; Fischer et al., 2012; Liu et al., 2021; e.g. Allen and Huggel, 2013) in a recently deglaciated area, to be assessed. The only

required input is an estimate of approximate landslide size and frequency for a particular spatial area. Sensitivity analysis could be undertaken to understand if variations in the magnitude-frequency relationship had a significant impact on the resulting risk estimates. It is also important to note that our risk analysis does not include cascading hazards, such as landslide dam formation and associated dam break floods as well as catastrophic glacier multi-phase mass movements, which may be important in an Alpine Fault earthquake scenario (Robinson and Davies, 2013). Such cascading hazards could be incorporated into future risk

analysis potentially using an event tree approach (e.g. Macciotta et al., 2016).

**5.3 Risk communication and management**

Our analysis quantitatively estimates the risk to life to visitors from landslides, with this information used by risk managers and decision makers to evaluate risk tolerability, determine appropriate risk mitigation measures and communicate the risk to visitors and workers in each valley. Due to the uncertainty associated with risk analysis (Lee and Jones, 2014), we report our

risk estimates as bands and not as single points (see Figure 10). The risk bands represent our central estimate through to upper estimate of the risk; we do not present a lower estimate of the risk as lower estimates are not currently used in decision-making regarding risk acceptability, in order to ensure that the highly uncertain risk levels are not underestimated. However, we note and agree with Strouth and McDougall (2021) that risk assessment conservatism should be avoided, with central estimates used for risk evaluation and uncertainties in the risk analysis presented transparently. The risk bands can be presented against

risk comparator data to inform risk evaluation and risk tolerability processes (cf. Taig, 2021). In Figure 10, the individual risk per trip for visitors to both Franz Josef and Fox Glacier Valleys (though it is important to note that the risk numbers do not include any mitigation measures and are therefore not residual risk), are plotted against other activities that a visitor may undertake. These activities include popular tourist activities in New Zealand, modes of transport to and from the Glacier Valleys, and risk per trip in other national park settings globally. More information on these datasets can be found in Taig

(2021 a, b). Not only can this be used to inform the risk evaluation process but can also help with risk communication to visitors. The range in risk values can be presented as a graphic to illustrate the risk, and to avoid confusion with small numbers or scientific notation along with helping visitors, whose main language may not be English, understand the uncertainty in risk results (Taig, 2021).

The disaggregation of the QRA allows a greater understanding of both the contributors to landslide risk, and their associated uncertainty. Such an approach presents a useful tool to inform and communicate to risk managers where appropriate management and mitigation strategies may be most effective. Reductions in vulnerability and exposure can be important risk mitigation measures (e.g. Schneiderbauer et al., 2017), as highlighted by our risk sensitivity scenario analysis. The LPR maps



can be used to inform track placement and re-alignment, reducing the time spent and exposure of an individual in high hazard
zones beneath steep slopes. The LPR maps may also be used for informing and identifying suitable, "less risky" stopping
points in the valley, when tracks are partially closed due to rainfall, high stream flows or other events. Where it is not possible
to relocate the tracks, other mitigation measures such as track closures during heavy rainfall may reduce the risk from rainfall-
induced landslides within the aseismic landslides class. As aseismic landslides dominate the risk profiles, reduction in exposure
to rainfall-induced landslides may result in a significant reduction in visitor risk per trip. However, due to the limitations of
our landslide inventory, we are unable to link landslide occurrence to rainfall events in each valley. To provide a robust basis
for using track closures as a risk reduction method, rockfall and landslide events should be documented and recorded within
each valley, along with meteorological observations. This should also include any information of the occurrence of debris
flows, particularly within Fox, as this landslide type is difficult to determine within our landslide inventory analysis and are
therefore underrepresented in our magnitude-frequency analysis even though debris fans are higher risk environments (see
Figure 6). A rockfall/landslide register can also be used to inform dynamic risk analysis, by recording areas of locally high
activity, with our methodology presenting a base risk model that can be easily updated and amended to incorporate future
information.


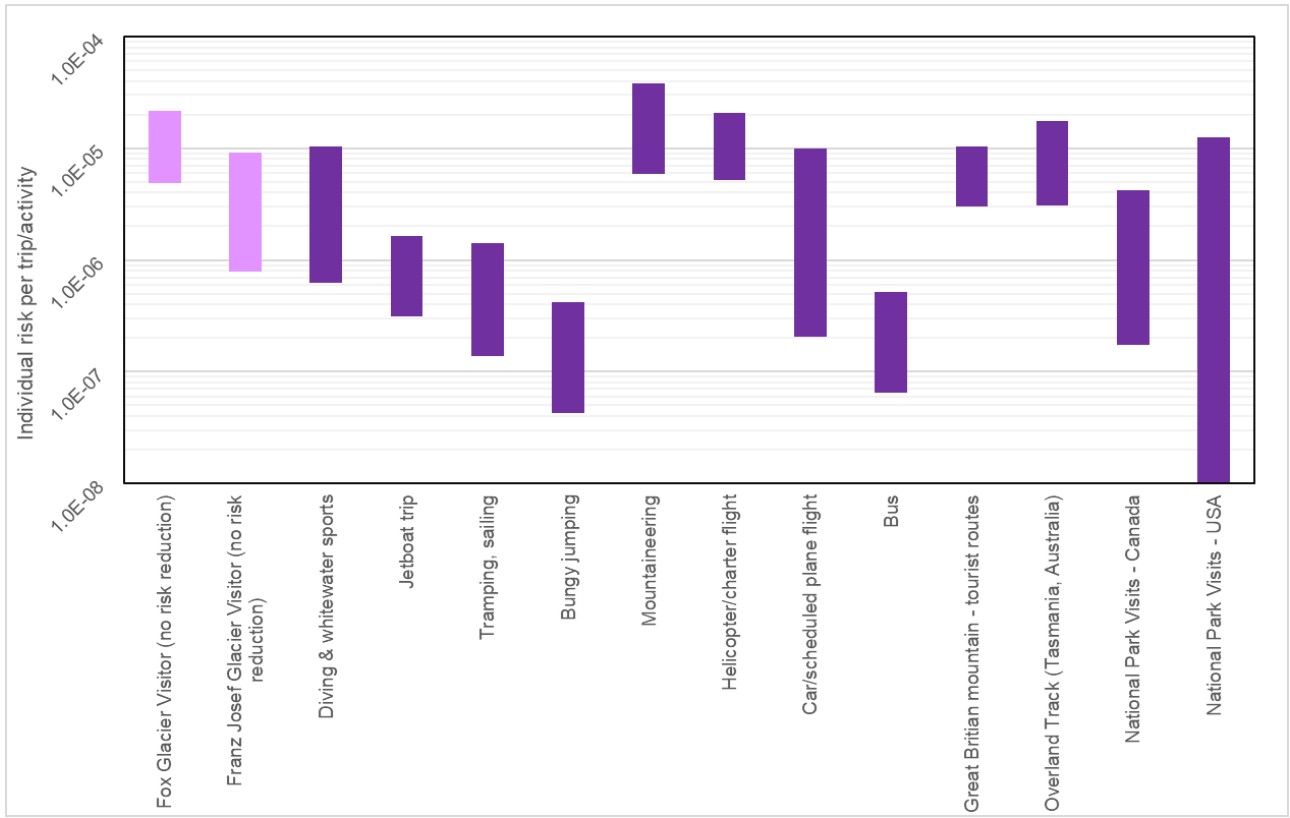

**Figure 10: Quantitative estimates of individual risk per trip for Fox Glacier and Franz Josef Glacier visitors compared against popular tourist activities, modes of transportation used to access the glaciers, and individual risk per trip to national parks overseas (sourced from Taig, 2021). The band range for each activity represents both the statistical uncertainty and uncertainty in the denominators of units of activity undertaken (cf. Taig, 2021).**

## 6 Conclusion

We have presented a quantitative risk analysis (QRA) case study from the Franz Josef and Fox Glacier Valleys, on the West Coast of the South Island, New Zealand. We deconstructed the QRA to reveal the relative contributions of aseismic versus seismic landsliding, and landslide volume classes to risk. Our results reveal that for both valleys in our central estimate scenario aseismic landslides contribute more to the overall risk than that of seismic landslides. However, our sensitivity analysis of nine risk scenarios, to explore the uncertainties in our inputs to the model, suggests that the contribution of seismic or aseismic landslide risk is dependent on time-variable input assumptions. The increasing probability of a large Alpine Fault earthquake occurring results in increased seismic landslide risk, both individual and societal. Increases in the number of aseismic landslides, within the standard deviation of the valley specific magnitude-frequency relationships, also increase the landslide risk, particularly in Franz Josef. This increase in aseismic landsliding may reflect climatically induced changes in landslide





rates in these actively deglaciating valleys and suggests that the risk of landsliding will change under different climate change scenarios. Additionally, the spatial location and susceptibility of landsliding may also change in response to environmental changes. We presented an example to show-case how local changes in the rates of landsliding can be explored and incorporated
in the analysis. We present our risk results as bands, not points, that display the uncertainty of our risk results. We suggest that QRA is not only a valuable tool for evaluating the risk to an individual but can be used to better understand what drives landslide risk and as such what risk management decisions will be most effective and appropriate in significantly reducing risk. In order to do this, QRA must be able to be deconstructed as well as be dynamic to account for changing hazards and exposure with time.






## Appendix

### Appendix A:  Aseismic landslide inventories

The data sources for the aseismic landslide include 1) a rockfall register compiled from observations made by staff of the
Department of Conservation (DOC), Franz Josef Glacier Guides Ltd, and Fox Glacier Guides Ltd, 2) a landslide inventory
derived from historical aerial imagery analysis of both valleys, and 3) a large landslide inventory of historical landslides
observed in the wider southern alps. The record of observed rockfall activity in the rockfall register contains data collected
since 2008 for Franz Josef and 2009 for Fox. The rockfall registers record the date, approximate size and source location, if
identifiable, of rockfalls. Alongside this, local knowledge of long-term guides Craig Buckland, Jon Tyler (Franz Josef Glacier
Guides Ltd) and Marius Bron (Fox Glacier Guides Ltd) informed the relative changing rates and sources of landslide activity
within each valley. We identified and mapped landslides from a series of historical aerial photographs for each valley (1948,
1965, 1981, 1985, 1987, 2011, 2017 in Franz Josef Glacier Valley, and 1953, 1981, 2017 for Fox Glacier Valley), and these
landslides were subsequently verified in the field. Additionally, we identified large (>500,000 m³) relict landslides in each
valley, with an unknown temporal occurrence. To assess the potential for large landslides to occur, we used a landslide dataset
recorded in the wider Southern Alps region, where there is evidence of large landslides occurring under aseismic conditions,
such as the 2007 11 million m³ Young River landslide (Massey et al., 2013). We also used data from the following studies
which detail the occurrence of debris avalanches since 1978: McSaveney (2002), Hancox et al. (2010), Cox and Allen (2009)
, Allen et al. (2011), Allen and Huggel (2013), Massey et al. (2013), and Cox et al. (2015). We assumed that landslides
<500,000 m³ are unlikely to have been noticed or mapped unless they impacted people or property in the wider Southern Alps.
We also assume that landslides in both valleys where the glacier guides and DOC operate would be well documented as people
are present almost on a daily basis.

### Appendix B: Aseismic landslide susceptibility models

We used best sub-set regressions to explore which group of variables could statistically best explain landslide occurrence.
From these variable groupings, we undertook backward step-wise regression modelling to determine which group of variables
was the most statistically significant. Using the variables of slope angle, local slope relief (LSR), material type and vegetation,
we estimated the aseismic landslide probability using the following logistic regression equation (with output coefficients in
Table B 1) for Fox study area:



$$\textbf{Non EQ LS prob} = \textbf{1}/(\textbf{1} + \textbf{e}^{(-(\textbf{Intercept}+\textbf{Slope Angle}+\textbf{LSR}+\textbf{Material type}+\textbf{veg}))})$$  (B1)

For the Franz Josef study area, we found vegetation to be a statistically insignificant variable when used to explain landslide

occurrence, and as such was not included in the model. For the aseismic landslide probability in the Franz Josef study area we

used the following logistic regression equation (with output coefficients in Table B 2):

$$\textbf{Non EQ LS prob} = \textbf{1}/(\textbf{1} + \textbf{e}^{(-(\textbf{Intercept}+\textbf{Slope Angle}+\textbf{LSR}+\textbf{Material type}))})$$  (B2)

**Table B 1 : Summary table of co-efficient estimates for the variables used in the Fox study area logistic regression equation.**

| Parameter Type | | Estimate (coefficients) | Standard error | Statistical significance (p) |
|---|---|---|---|---|
| Intercept | | -7.07532 | 0.235784 | 0.000000 |
| Slope Angle | | 0.03376 | 0.000177 | 0.000000 |
| Local Slope Relief | | 0.02067 | 0.000188 | 0.000000 |
| Material Type | Rock | 1.33131 | 0.235734 | 0.000000 |
| | Alluvium | -5.16420 | 0.471425 | 0.000000 |
| | Colluvium | 0* | NA | NA |
| Vegetation | Vegetated | -1.08698 | 0.003315 | 0.000000 |
| | No Vegetation | 0** | NA | NA |

*Colluvium is set as the reference material, which means that Alluvium is less likely to fail with a negative estimate and rock more likely to fail with a positive estimate.

**Slopes that are not vegetated is set as the reference vegetation variable, which means that areas that are vegetated are less likely to fail with a negative estimate.


**Table B 2: Summary table of co-efficient estimates for the variables used in the Franz Josef study area logistic regression equation.**

| Parameter Type | | Estimate (coefficients) | Standard error | Statistical significance (p) |
|---|---|---|---|---|
| Intercept | | -5.70152 | 0.061317 | 0.000000 |
| Slope Angle | | 0.01196 | 0.000176 | 0.000000 |
| Local Slope Relief | | 0.00506 | 0.000159 | 0.000000 |
| Material type | Colluvium | 1.33545 | 0.061123 | 0.000000 |
| | Rock | 2.39511 | 0.060985 | 0.000000 |
| | Alluvium | 0* | NA | NA |

**Alluvium is set as the reference material, which means that colluvium and rock are more likely to fail with a positive estimate.



**Appendix C: Landslide runout analysis**

**C.1 Rockfall**

We modelled landslides with volumes ≤1,000 m³ as rockfalls using RAMMS rockfall software (2015) for all material types. The software simulates the rigid body motion of falling rocks and predicts rock trajectories in general three-dimensional terrain. Rock trajectories are governed by the interaction between the rock, its associated shape, and the nature of the ground (e.g. a soft substrate such as sand will dampen the rock energy in contrast to a hard substrate such as rock). Generalised rock shapes

are simulated, and rock block orientation and rotational speed are included in the rock/ground interaction. We determined the simulation parameters for forecasting by back-analysing recorded rockfalls within the study areas, where the source area, boulder shape and rockfall trails were recorded or could be accurately inferred. The RAMMS rockfall forecast parameters adopted from back analysis are shown in Table C 1, along with descriptions of the parameters and the data sources used to derive them. The results from the simulations comprise: kinetic energy; runout distance; jump heights and the number of

simulated trajectories passing through a given grid cell.





**Table C 1: RAMMS Rockfall model parameters used for forecasting rockfalls.**

| Simulation Variable | Description | RAMMS parameter | Data source |
|---|---|---|---|
| Substrate material | Alluvium, swamp | Terrain parameter: Soft | Materials taken from the engineering geomorphology materials layer |
| | Colluvium, talus and moraine, and mixed colluvium, moraine and talus | Terrain parameter: Medium | |
| | Rock, rock at/near surface | Terrain parameter: Hard | |
| Vegetation | Scrub | Forest parameter: Open forest | Mapped from aerial photographs and field verified |
| | Trees | Forest parameter: Medium forest | |
| Rock shape | The shape of the boulders used in the simulations | Rock parameter: "Real long", dimensions (1.5 by 1.0 by 1.0 m). Rock volume = 1 m$^3$ (assumes rounded edges). Mass = 2,730 kg | Field mapping and measurements of rockfalls |
| Topography | The digital elevation model used in the simulations | Terrain = 3 m by 3 m grid cell resolution | Digital Elevation Models (DEM) (bare earth) derived from the LiDAR surveys of both study areas |
| Release | Number of random orientations of the rock blocks at source | Three random orientation were selected | N/A |
| | Source area locations | Rock positions: from 3 m by 3 m grid cells, with slope angles ≥45º | From the LIDAR DEMs |
| | Initial velocities of the rock blocks | Initial velocities of: X = 1.5 m/s, Y = 1.5 m/s and Z = 1.0 m/s, were assumed | N/A |

## C.2 Debris avalanche and flows

To identify the areas impacted by landslides, and the associated height of debris and number of boulders, we conducted a suite of runout simulations for the different volume bins. We modelled landslides with source volumes > 1,000 m³ as debris avalanches (if sourced from rock) or as debris flows (if sourced from colluvium or moraine) using RAMMS debris flow software (2011). RAMMS is based on Voellmy friction law, where the frictional resistance consists of a dry-Coulomb type friction (coefficient μ or Mu), which scales with normal stress, and a viscous turbulent friction (coefficient xi), which scales with landslide volume. These coefficients are calibrated from the back-analysis of case-studies. For this assessment, we used


the back -analysis of 67 debris avalanches (ranging in volume from 300 m³ to 100 Mm³) published in the literature (Schneider et al., 2011; Allen et al., 2009). For debris flows, we used 22 back analysis case-studies ranging in volume from 1,000 m³ to 200,000 m³ (Loup et al., 2012; Cesca and Agostino, 2008; Deubelbeiss et al., 2011; Hussin, 2011; Scheuner et al., 2011). We

fitted a power-law to the data (Figure C1 and Figure C2) to calculate the coefficients for the numerical simulations. For debris flows, the Xi parameters did not vary with source volume and so we adopted a central estimate of 350 in the numerical simulations.

In areas where the source area could potentially fail as either a debris avalanche or debris flow (for example, potential failure from the top of the larger creeping Yellow Creek landslide in Fox study area), we simulated both and calculated the maximum

debris flow height from the two outputs.

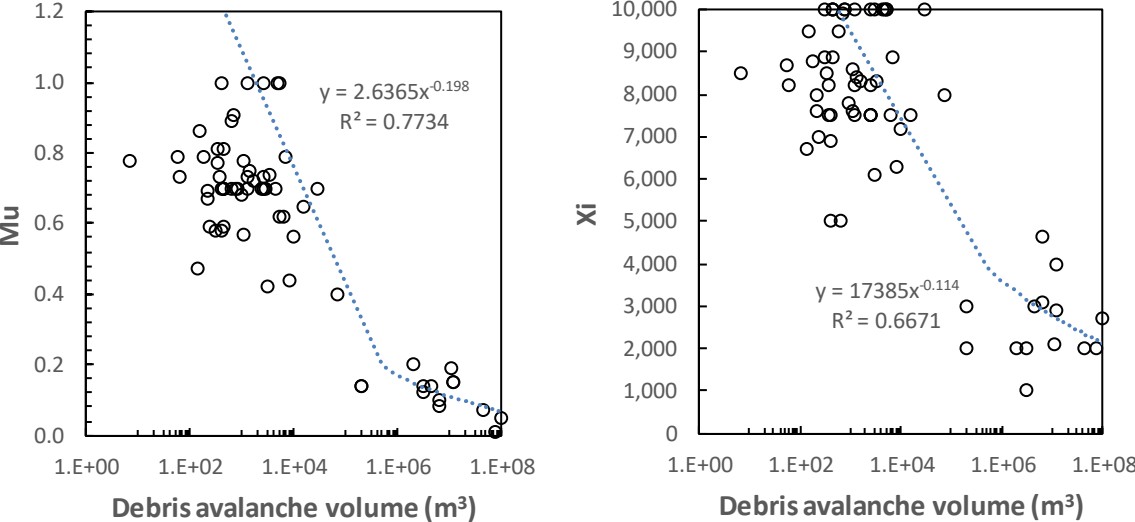

**Figure C 1: Debris avalanches: Range of parameters used to back-analyse the runout of debris avalanches published in the literature (n = 67), using the RAMMS software.**



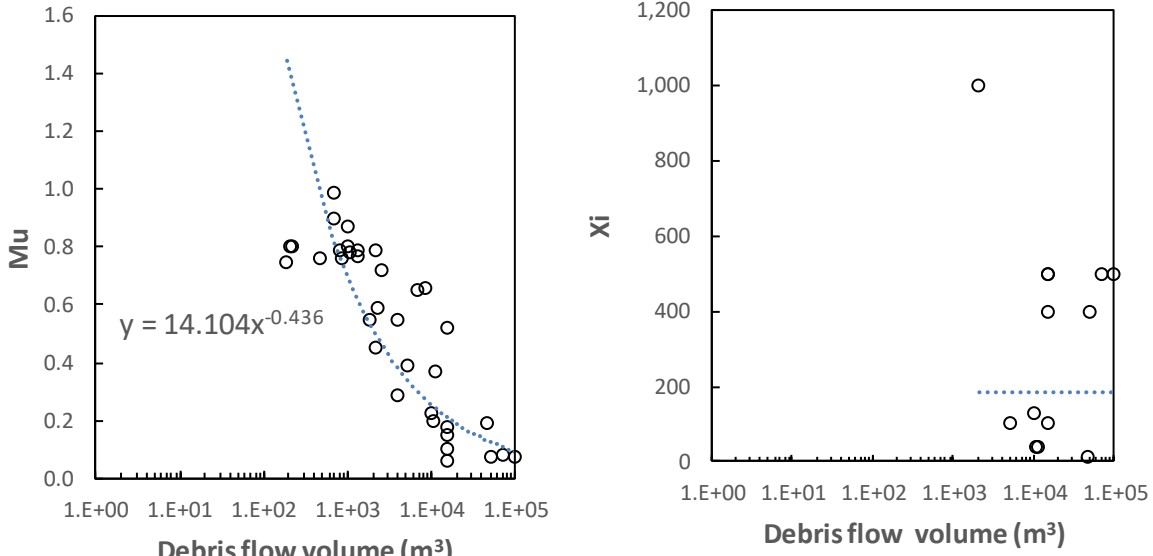


**Figure C 2: Debris flow: Range of parameters used to back-analyse the runout of debris flows published in the literature (n = 22), using the RAMMS software**

**C 3: Sensitivity analysis of debris avalanche and debris flow output**

We assessed the sensitivity of the simulated maximum debris heights to varying RAMMS input parameters. Different input Mu and Xi parameters within RAMMS result in a change in both the extent of the debris runout (and therefore area inundated by debris) and the height of the debris and therefore the number of boulders passing through a given location on the ground (grid cell). For the debris avalanche simulations, we calculated the standard error of the modelled fit of the data (Figure C. 3) We both added and subtracted the standard error from the power-law relationship to obtain the mean ±1 standard error (SE)

values of both Mu (µ) and Xi parameters (Figure C3). For the debris flow simulations, we used the same procedure as outlined for debris avalanches to calculate the Mu parameter. As no relationship existed for the Xi debris flow parameters (Figure C 2), we used the standard deviation (σ) of the mean parameters to calculate both mean +1σ and mean -1σ ( Figure C4). We choose one representative source area for both debris flow and debris avalanche deposits, to simulate both mean +1 SE (or standard deviation) and mean -1 SE (or standard deviation) runouts. For each source area, the following volume classes were simulated;

10,000 m³, 50,000m³, 100,000 m³ and 1 million m³. We varied the simulated volume size to assess if the range in maximum flow height increased or decreased for larger volumes. From the simulation results, we calculated the difference in debris heights (per grid cell) between the standard parameter simulation results and the results from each respective mean ± 1 SE simulation. We calculated the mean and standard deviation of the difference (in debris height per grid cell) for each simulation result (i.e. we calculated the difference of the difference). We summed the values to calculate the mean +1σ (upper-bound)

value of the difference between the simulations.





The results from our sensitivity assessment indicate that the absolute difference in debris heights increases with volume size, whereby larger landslides can display several metres of difference in flow height for any given grid cell. Proportionally, the differences in maximum flow height for debris avalanches were on average 60% ± 22% higher than those modelled using the preferred forecast parameters. For debris flows, the difference in maximum flow heights are on average 60% ± 28% higher

than those simulations adopting the preferred forecast parameters. We applied this 60% factor of difference to all simulation results to derive upper estimates of debris heights.

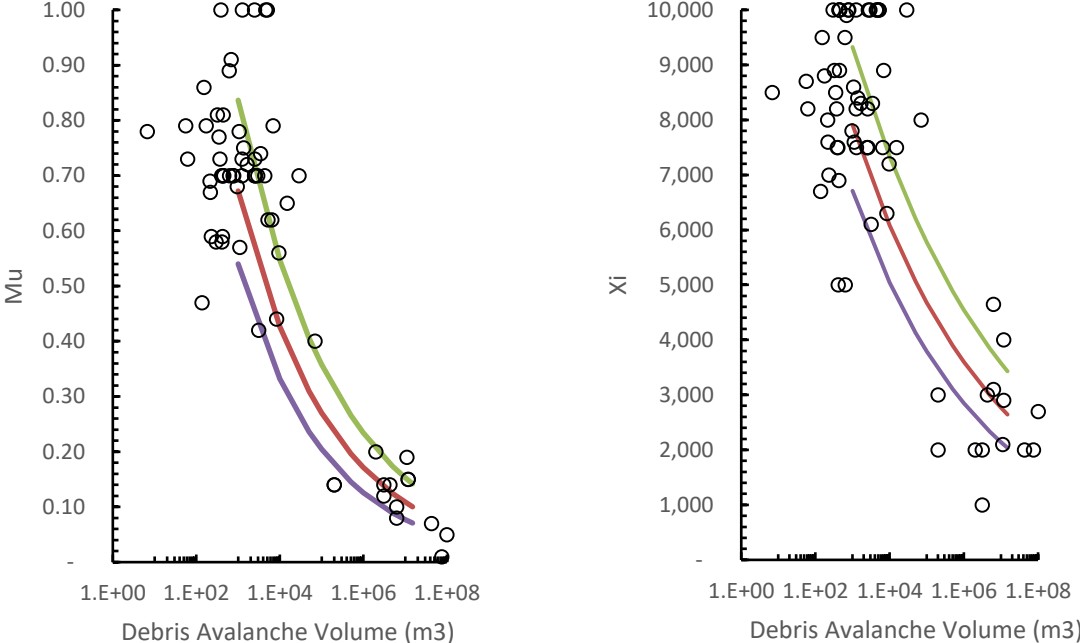

**Figure C 3: Range of parameters used to back analyse the runout of derbis avalanches published in the literature. Purple fitted line represents the mean -1σ, the red line represents the modelled fit of the data, and the green line represents the mean +1σ.**




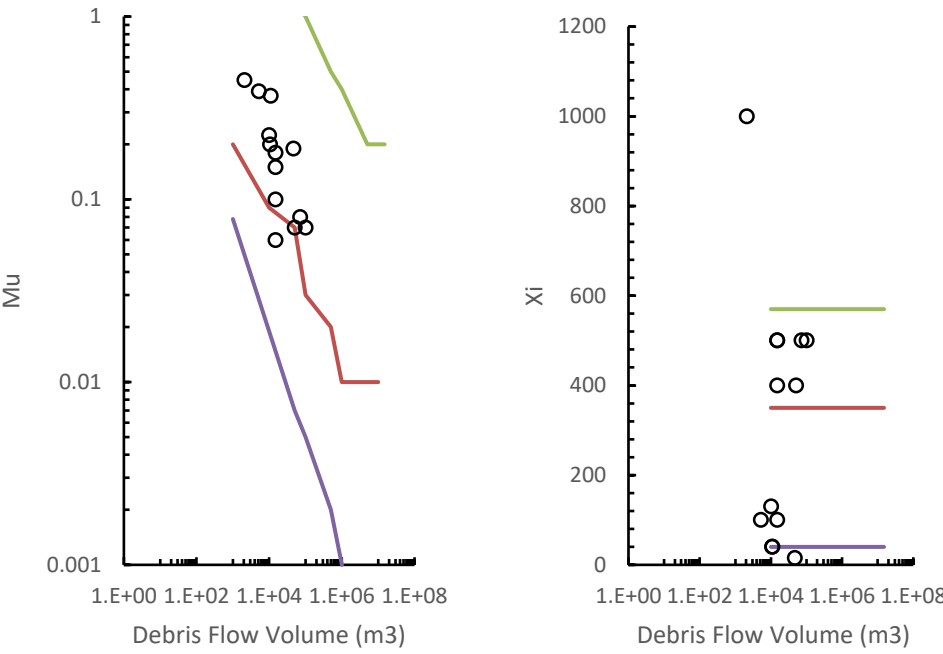

**Figure C 4:Range of parameters used to back analyse the runout of derbis flows published in the literatures. The purple fitted line represents the mean -1σ, the red line represents the rounded parameters used within the simulation, and the green line represents the mean +1σ.**

**Appendix D Calculation of LPR**

For seismic landslides, the probability of death ($P_D$) is calculated for each earthquake Band 1 to 4, for each individual source area of a given volume class, and the debris that the given source area generates. The calculations were done for each grid cell within the study areas. Firstly, we calculated the probability of death for each landslide source and its related debris ($P_{D(Source)}$), where:

$$P_{D \text{ (EQ Band x; Vol Class y; Source z)}} = P_{A \text{ or B (dependent on LS Vol Class)}} \times P_{2(S:H)} \times V \qquad (D1)$$


$P_{A/B}$ is the probability the given source within each landslide volume class, will generate the given volume of debris. $P_2$ is the probability of being in the path of N boulders within the debris generated by a given source, if it occurs. V is the vulnerability, defined as the probability of a person being killed if present and in the path of one or more boulders. We then calculated the $P_{D(Source)}$ for each source area of a given landslide volume class (y), and its related debris, and combined for each landslide

volume class to estimate the probability of death from all landslides of the same volume class that might contribute to the given grid cell ($P_{D(Vol\ Class)}$), where:





$$P_{D \text{ (EQ Band X:Vol Class y)}} = 1 - \left(1 - P_{D(\text{Vol Class y1; LS Source z1})}\right) \times \left(1 - P_{D(\text{Vol Class y1; LS Source z2})}\right) \times \left(1 - P_{D(\text{Vol Class y1; LS Source zN}...)}\right) \tag{D2}$$

We then calculated the probability of death from ALL landslides of a given volume class generated by each earthquake band. $P_{D(\text{EQ Band x; Vol Class y})}$ for a given earthquake band (x) and volume class (y), was multiplied by the number of landslides of a given volume class ($N_{LS}$)) generated by the representative earthquake PGA of the given band. If the number of landslides ($N_{LS}$) triggered in the band was $\geq 1$, then instead of multiplying $P_{D(\text{EQ Band x; Vol Class y})}$ by the number of landslides ($N_{LS}$), the following formula was used:

$$P_{D(\text{EQ Band x; ALL Vol Class y})} = 1 - \left(1 - P_{D(\text{EQ Band x:Vol Class y})}\right)^{N_{LS}} \tag{D3}$$

We combined the contribution to each grid cell from each landslide volume class per band to calculate the probability of death from all landslides triggered by the representative PGA in the given band ($P_{D(\text{EQ Band x})}$), where:

$$P_{D(\text{EQ Band x})} = 1 - \left(1 - P_{D: \text{ ALL Vol Class 1k}}\right) \times \left(1 - P_{D:\text{All Vol Class 10k}}\right) \times \left(1 - P_{D:\text{All Vol Class N}....}\right) \tag{D4}$$


We calculated the local personal risk from all landslides that occur within the given band by multiplying $P_{D(\text{EQ Band x})}$ by the annual frequency of the presentative earthquake PGA in that band.

For aseismic landslides, we calculated the probability of death ($P_{D \text{ (Vol Class y)}}$) in the same way as earthquakes except ignored the need to calculate for each earthquake band. We then calculated the LPR for each volume class, by multiplying $P_{D \text{ (Vol Class y)}}$ by the annual frequency of the given volume class (y) of landslide occurring.


### 7.5 Mill's creek catchment magnitude – frequency relationship

We used several datasets to derive the magnitude-frequency relationship for the Mill's Creek Catchment, including: 1) A change detection model from differencing of a March 2017 digital surface model (DSM) and June 2018 digital elevation model (DEM); 2) National park staff observations of the frequency of debris flow events and a rough estimate of their associated 745 volume; and 3) NIWA weather observation data from Franz Josef township, which represents the closest meteorological observation point. Our change detection model revealed that between March 2017 and June 2018, approximately 6.5 million m³ was eroded and 3 million m³ deposited within the Alpine Gardens and Mill's Creek catchment. During this same time period, the valley was closed 34 times due to heavy rain and flooding. For the larger storm events, including ex-tropical Cyclone Fehi in February 2018, national park staff observed debris flow activity that resulted in damage to the road. The staff 750 estimated that for the Cyclone Fehi event, approximately 2 million m³ had been deposited on the Mill's Creek debris fan (Tony Hart – personal communication). Using this information, including both the rough national park staff volume estimates and frequency of heavy rain events likely to trigger debris flows, we divided the approximately 6.5 million m³ into different debris

flow events based on magnitude – frequency principles. We normalised the data over the 1.38 year time record and the spatial area of the Alpine Gardens and Mills Creek catchment to derive a magnitude frequency power – law relationship, with elevated

rates of landslide activity compared to the magnitude – frequency relationship for the Fox Glacier Valley overall.

**Author contribution**

SdV wrote the manuscript, collated data inputs, conducted hazard and risk analysis. CIM contributed to the manuscript and conducted hazard and risk analysis. BL contributed to the manuscript and conducted susceptibility and risk analysis modelling. TT contributed to the manuscript plus provided input into the risk analysis, evaluation, and management. RM contributed to

the manuscript, collated data inputs and undertook runout modelling. SdV, CIM and RM undertook fieldwork towards the project.

**Competing interests**

The authors declare that they have no conflict of interest.

**Acknowledgements**

Funding and support for this project was provided by the Department of Conservation (DOC), New Zealand and the Strategic Science Investment Fund (SSIF), NZ. We would like to thank Wayne Costello, Owen Kilgour, Tony Hart and Don Bogie (DOC) for their help and support of the project. Thanks to Marius Bron (Fox Glacier Guides), Craig Buckland, and Jon Tyler (Franz Josef Glacier Guides) for their local hazard information and knowledge. Thanks also to Garth Archibald, Katie Jones, and Salman Ashraf (GNS Science) for help and support with fieldwork and remote sensing data.

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
