# Peer review of "What drives landslide risk: Disaggregating risk analyses, an example from the Franz Josef and Fox Glacier Valleys, New Zealand"

_Natural Hazards and Earth System Sciences, 2022_

## Author Comment (AC1)

**Response to Reviewer 1**

We thank Nicolas Barth for the helpful suggestions and comments, as well as taking the time to read and evaluate our manuscript. Please find outlined below our response to the main suggestions made on the manuscript, as well as our line by line response.

(1) Figures. At least one figure panel is mislabeled. Changes/additions to labeling and color ramps could improve readability and understanding. I suggest one possible figure that could be added and the addition of another panel to an existing figure. Detailed suggestions are below.

Thanks for both this main comment and the associated line comments. We will do a thorough check and review for all figures, figure captions, and labels to make sure they are clear and readable. This includes changes to the colour map ramp in Figures 5,6 and 9 to make sure that the detail in the order of magnitude changes in risk values are more obvious. We will also include a new Figure 2 in the Study Site section to explain the different landslide types considered within our risk analysis. We include information on the types of infrastructure in our amendments to Figure 1.

(2) References. Some cited references appear missing, notably Taig 2021a&b. I thought the landslide risk assessment introduction was generally well referenced but that there could be more background references on the range of landslides (historic+prehistoric) documented in the western Southern Alps environment (Korup, Hancox, etc.) and as far as I could tell the landslide inventory used is not specifically referenced and undoubtedly must draw from or build upon existing compilations (Korup, QMAP, etc.). I also suggest Yosemite Valley references for the discussion- they might not have the same range of events as Franz/Fox but certainly lots of parallels with rockfall risk to tourists on trails.

The Taig reports have now been finalised, so the reference will be updated and included in the revised manuscript. It is our understanding that DOC intends to make these reports available on their website. The reports provide the background for risk comparison dataset and the risk threshold advice for DOC. The references are:

Taig T. 2022a. Risk comparisons for DOC visitors and staff. Cheshire (GB): TTAC Ltd. 131 p.

Taig T. 2022b. Guidelines for DOC on dealing with natural hazard risk. Cheshire (GB): TTAC Ltd. 91 p.

The references for our aseismic landslide inventory are included in the Appendices. We included in the landslide inventory only historic landslides where we knew the timing of failure (e.g. it was a single failure event with a well constrained volume). This detail was moved to the Appendices to provide a more concise methodology section; however, we are open to adding this back into the main text if it better supports our aseismic landslide magnitude – frequency relationship. We will include more information (and associated references) on the range of wider southern alps landslides within the Study Site (Section 2) to set the scene for the type and size of landslides that could occur.

Within the 'Drivers of Risk' Section 5.1 of the discussion, we can include a discussion on prehistoric landslides, our assumptions in the magnitude-frequency relationships, and the associated uncertainty of the aseismic landslides. This further emphasises the need for robust landslide inventories.

Thanks for highlighting the Yosemite rockfall risk references. Their risk analysis highlights how effective reduction in exposure can be to reduce the risk. We will add these references in to appropriate sections of the discussion. This also includes the note to crowd-sourcing the information on landslide occurrence. The QR code on signage sounds like a great idea to explore.

*Line comments:*

L 23: Since your study areas top out at 2000m and go down to 200m it might be more appropriate to point to "high relief mountain areas" rather than "high mountain areas" (most of the affected areas are closer to 200m).

We will make the suggested change.

L62-63: Surely you can get pre-COVID annual visitation estimates and typical daily number of workers in each valley from DOC? Similarly a peak tourist season daily count of tourists and workers? I think that would help a lot with the context.

Yes, we do have estimates of annual and peak visitor numbers, so will include these in the revised manuscript. For Fox Glacier Valley, the annual pre-COVID visitor numbers are approx. 400,000 per year with a maximum number of visitor's trips into the valley of approx. 3,500, while Franz Josef received approx. 700,000 per year with a maximum number of visitor's trips into the valley of approx. 6,000.

L70: I prefer capitalization of the formal fault name "Alpine Fault" as you use elsewhere.

We will make the suggested change.

L74-75: A reference or two would be good here if possible. And actually the next two sentences too.

We will add in the appropriate references.

L87-88: "respectively" not needed.

We will make the suggested change.

L 540: Taig 2021 (a&b?) is cited multiple times as a seemingly important reference but does not occur in the reference list.

We will update the reference list, as mentioned above.

L560-567: Not required if you think it breaks your narrative but there are several good references to rockfall risk in Yosemite Valley (4M visitors/yr) including their extensive rockfall database that visitor are encouraged to submit observations to (crowdsourced). A QR code to report a rockfall/etc could be placed on the signage suggested at L546.

https://www.nps.gov/yose/learn/nature/rockfall.htm

https://pubs.usgs.gov/ds/746/

I agree with you that better documentation of events will go the furthest for robust decision making.

The appendix was skimmed but appears to be appropriate.

To better acquaint your readers with the types of hazards and potential exposure, it might be helpful to have a multi-panel figure showing photographs (aerial and ground) of different hazards (rockfall, debris flow, rock avalanche, etc) and the style of infrastructure (trails, roads, etc.) within the valleys. I would suggest this could be a good Figure 2 (sliding current Figure 2 to the 3 spot). And relatedly some text in the Study Site section.

See our response above with regards to both the Yosemite references and new Figure 2 and related text.

L132: Just a general comment that I follow the life risk calculation approach and am fine with it. As you point out obviously a lot of unknowns that lead to orders of magnitude ranges of possibilities. I find the vulnerability factor particularly interesting. I wonder if later on in the discussion it is worth considering a 100% vulnerable scenario (i.e. approximate scenario that a trail closes because a large landslide is witnessed, near miss, or someone is struck by a rock but fine)- potentially important from an economic standpoint.

Agreed that there is more to explore with regards to different scenarios. We suggest that our risk model provides the base to explore these different scenarios using e.g. event trees.  We will include such a point in our discussion section.

Figure 1: Nice figure overall. Some of the details are hard to see because of the resolution and jpg compression- I suggest the final version have a higher resolution. Maybe shade the glaciers in transparent blue polygons to improve context? Show Alpine Fault in Panel B? Maybe label bridge locations at outlets of the study areas? Clipped study area boundaries are a little arbitrary, especially Fox case study that crops the unnamed west-draining creek off Pt 1401, Serac Creek, and unnamed west draining creek from Mt Garnier (but whatever I guess).

We will make the suggested changes, including adding in the Alpine Fault to Panel B. Will include information on the infrastructure (road, tracks, bridges) on Panel C and D. We will update the Fox case study area boundary to better reflect the areas of our analysis, as we included and ran runout models from both the unnamed west -draining creek off Pt 1401 and the unnamed west draining creek from Mt Garnier.

Figure 2: Maybe add the PGA labels to the "Band 2"/etc in the legend so the figure stands alone better.

We will make the suggested change.

Figure 3: Slope angle legend is wrong in panel B (flipped). Specify how the local slope relief in panel C was created (typically a radius of a chosen width?). Provide citation for

data in panel A and panel D. What is the source and basis of the vegetation mapping in panel E? The bedrock ridge of Cone Rock (high probability in panel F) should be vegetated.

We will modify Panel B. We created the Local Slope Relief using a radius of 80 m based off the methodology in Massey et al., 2018. We used both Landcare's Land cover database and aerial photography collected in 2017 and 2018 DEM as the basis for the vegetation classification. We will add this information into the text and figure caption regarding this. With regards to Cone Rock, some of this area should have been classified as vegetated, though the high probabilities in Panel F aligned with either pre-existing rock slope failures or steeper with limited vegetation rock slopes, and as such we do not think that this will have a large impact on the results.

Figure 5&6: Pretty much same comments for both figures. The color ramp in panel A is so smooth (17 shades of blue) such that visually you can probably only determine color values to +/-1-2. It looks nice but a more variable color ramp would convey more information. In panel C it is hard to differentiate a lot of the colors chosen, particularly 10m3 and 10000m3. In panel D again I think helpful to have PGA ranges in legend.

We will make the suggested change to the colour ramp in both Panel A's. Will try and make the lines easier to differentiate in Panel C, as well as adding in the PGA range in the legend.

Figure 7: Is it worth scaling the two panels to the same LPR scale? Maybe some loss of detail but would help in comparing the two valley tracks directly.

We can make the suggested change – agreed that this will help with comparison between the two valleys.

Figure 8: Nice. Maybe mark the track positions on the different images? Was the Chalet Lookout Track marked on the LINZ topo abandoned due to this landslide?- dot that in to further illustrate the point?

Yes, the Chalet Lookout Track was abandoned fairly quickly after the landslide initiated, so will add the track positions onto the different panels.

Figure 9: Suggest a more variable color ramp again. Just a comment that it is interesting to see the SE of Cone Rock spillover in the LPR- a nice argument for the value of high res topo data in these analyses.

Will make the suggested change in colour ramp. Agreed with regards to the high-resolution topography, which is confirmed with a couple of debris flow events observed spilling over in the SE Cone Rock location.

Figure 10: I found this a really useful figure to help contextualize the results of your study, particularly in a way readers and policy makers can understand. While the per trip metric is helpful for evaluating tourist's risks, it is less helpful for evaluating the risk of workers who have prolonged and multiple exposures. It would be helpful to be able to evaluate the

risk of workers in the valley (trail maintenance, road worker, tour guide, etc.) compared to other places in New Zealand alongside this "one visit" risk. Presumably someone doing trail maintenance will linger in the more dangerous areas longer. Seems like you may have a lot of these data in Massey et al. 2018c? Wishful thinking perhaps but it would make a nice second panel to this figure.

There is companion figure in Massey et al., 2018c that shows the risk comparators for DOC workers (who walk the tracks daily) compared to other workplaces in NZ. Additionally, Taig (2022 a & b) provides guidance for DOC on determining risk to workers who are exposed to landslide hazards regularly versus workers that may only be temporarily exposed for short time frames.

To keep the focus of this paper on visitor risk and make sure that the length of this paper is appropriate, we are not likely to include a second panel on worker comparator risk. Though as with the Taig reports, it is our understanding that the Massey et al., 2018c report will be made available on the DOC website.

---

## Author Comment (AC2)

We thank Tom Robinson for the helpful suggestions and comments, as well as taking the time to read and evaluate our manuscript. Please find outlined below our response to the suggestions made on the manuscript. We hope that we have added clarity where needed to the text. We have not transferred information from the appendix in the main text, as in response to the comments below additional information (not found within the appendix) was required to address the comments.

**Introduction:**

A few more key references on landslides in S Alps would be useful – Korup, Davies, McSaveney etc all have plenty of articles relevant here that would be useful background.

We can include links and references to landslides in the Southern Alps in line 60:

*"In the Southern Alps of New Zealand, landslides are a common feature that play a significant role in driving erosion (e.g. Hovius et al., 1997, Korup et al., 2004) and present an increasing natural hazard and risk to people ad property (Allen et al., 2010, Cox et al., 2015; McSaveney, 2002)."*

It would also be good to see some more landslide QRA works referenced, at least breifly as there are certainly several around that would be useful to highlight

We can include a sentence to highlight the different uses and applications of landslide QRA in Line 33:

*"QRA are undertaken for a land use planning (e.g. Bell and Glade, 2004, Vega and Hidalgo,2017), infrastructure (e.g. Voumard et al., 2013. Macciotta et al., 2015), and for visitor destinations ( e.g. Corominas et al., 2019; Stock et al., 2014)."*

**Study Site**

This needs an overview of at least the pre-covid number of visitors for reference. How many people on average visit per day?

We will include these numbers in the revised manuscript. For Fox Glacier Valley, the annual pre-COVID visitor numbers are approx. 400,000 per year with a maximum number of visitor's trips into the valley per day of approx. 3,500, while Franz Josef received approx. 700,000 per year with a maximum number of visitor's trips into the valley per day of approx. 6,000.

L75: Aseismic landslides needs a reference to support

We will add in the appropriate references.

**Method**

Each representative earthquake event? Some more details on this would be good – is this just an Alpine Fault event or does this consider far-field sources too? What about potential seismic sources within the ranges (e.g. Cox et al 2012 – Tectonics)?

We will add more detail into the paragraph starting at Line 200, around the details of the national seismic hazard model and what it accounts for. In the NSHM, the active fault component defines the Alpine Fault local to Franz Josef as the AlpineF2K fault source. Within the NSHM the AlpineF2K source generates a $M_w$ 8.1 ± 0.2 earthquake with a single-event (strike-slip + dip-slip) displacement of c. 9.2 m with a mean recurrence interval of 341 years (Stirling et al., 2012). This is time independent variable and does not consider time elapsed since the last earthquake on the Alpine fault in 1717. Landgride et al., (2016) deaggregated the NSHM to see what other fault sources may contribute to the shaking hazard at Franz Josef. For a probability of roughly 10% in 250 years (or 2,500 years) the deaggregation indicates that the main contributor of seismic hazard is the $M_W$ 8.1 AlpineF2K source (i.e., the Alpine Fault). Additionally, the second largest seismic hazard over 2,500 years comes from moderate magnitude ($M_W$ 5-6) earthquakes that can occur <10 km from the townships. Although the Alpine Fault is the main seismic source in the area, the section of fault that could rupture might be located some distance away from the sites. For this reason, and to consider the contribution from the $M_W$ 5-6 earthquakes, we, therefore estimate the landslide severity for the four different bands of PGA as determined from the NHSM.

Compiled info on visitor / worker duration – could you expand this description a little here in the main body. This is crucial to understand some of the key variables to the risk equation. You've provided some nice details for the hazard part, so it would be good here to have some details on the exposure part. For instance, is this data pre-covid (something for the discussion). Is it averaged, or do you take demographics into account which may change exposure time (e.g. how did you determine an average walker vs a slow one?).

We will add more detail to Section 3.6. Our walking times for an average and slow walker was determined from DOC data and estimates. In Fox Glacier Valley, our average walker spent 1.5 hours walking to and from the glacier viewpoint and 0.2 hours driving to and from the car park (see Figure 1 c), while the slower walker spent 2 hours walking to and from the glacier viewpoint and 0.3 hours driving to and from the car park. In Franz Josef Glacier Valley, our average walker spent 2-hour walking to and from the glacier view point, and 0.3 hours driving to and from the car park (see Figure 1 d) while the slower walker spent 2.5 hours walking to and from the glacier viewpoint and 0.4 hours driving to and from the car park.

We also conducted our own field counts and checks on walking speed and approx. visitor numbers. This also revealed that not all visitors travelled the full length of the tracks and turned back at certain points. More detailed analysis and investigation of visitor behaviour and its impact on both exposure and vulnerability would be interesting to include but outside the scope of this study.

For the societal risk calculations (not discussed within this paper) we used pre-covid data of visitor numbers. Additionally, since 2019 and 2020 the access into Fox and Franz Josef valleys respectively, has been reduced having knock on effects to both individual visitor risk and societal risk.

Empirical estimates of vulnerability – largely agree, although the central estimate for 1000 m3 seems optimistic to me, even with evasive action

It is an average estimate for all landslide volumes below 1000 m3, as a 1000 m3 represents the bin boundary. As such, evasive action may contribute more to the vulnerability at lower volumes than at higher volumes within the bin, and as such for the central estimate is a representative value. However, we do account for the uncertainty by assuming a value of 1 in our upper estimate.

Seismic landslide inventories – are these 3 events likely to be representative though – rock types are similar enough as is the topography, but the climate is variable as is the earthquake history. Perhaps a point for the discussion, rather than expanding here in the methods but worthwhile all the same

We will include a point about this in the discussion under Section 5.1 regarding the importance of landslide inventories and uncertainties associated with the use of our 3 earthquake landslide inventories (see below).

*"For seismic landslides, the landslide inventories of the 2016 $M_W$ 7.8 Kaikoura, 1968 $M_W$ 7.1 Inangahua and 1929 $M_W$ 7.8 Murchison earthquakes (Massey et al., 2018b; Hancox et al., 2014, 2015), were used as proxies for Franz Josef and Fox Glacier Valleys given the lack of seismic landslide inventories for the West Coast. All three inventories were dominated by shallow debris avalanches, with such failure types potentially being the dominant type of seismic landslide type (Keefer, 2002). The schist rock mass of both glacier valleys is fractured with persistent faulting (Cox and Barrell, 2007) and therefore we assume that shallow debris avalanches are the dominant failure type. While all three inventories occurring in similar mountainous terrain to Franz Josef and Fox Glacier Valleys, climatic differences exist, with the impact of these climatic differences on the number and size of seismic landslides triggered unknown."*

L204 – Alpine Fault earthquake date needs a reference

Will add in the appropriate reference.

You've assumed landslides won't occur on slopes <30 deg – could you not use your compiled inventory to assess just how likely this is? Surely you have a slope frequency distribution you could use to inform this decision, or at least weight the probability component?

Our analysis of our landslide inventory shows that no landslides occurred on slopes less than <30 degrees. We will add this detail into Line 231. Slope angle is used within our susceptibility models to weight the probability component of the risk model.

Fig 2 – hard to read the legends and quoted power laws, particular in panel a

We will adjust to figures to make them more legible.

PGA input from NSHM – does this vary much over the valleys or is it pretty constant? Given the short valley lengths and distance from the Alpine Fault I would have thought there is little variation across the valleys, meaning it's the other factors that play the biggest role in determining landslide source?

The PGA input varies from 0.8 g to 1.1 g across both valleys. The other components of the earthquake induced landslide susceptibility model of Massey et al., 2018, such as distance to fault, slope angle, geology and local slope relief therefore also dictate landslide source probability.

Fig 3 – would be good to see the NSHM here to since that's a key input for the seismic landslides

We can add in a box for the NSHM and the output earthquake induced landslide susceptibility model.

All slopes >45deg can generate rockfall – I don't necessarily disagree, but what is the justification for this?

We will add in the appropriate reference.

Field measurements show average boulder size is 1 m3 – again, would be good to see the distribution of this here in the main text somewhere to help support this - it would also be useful to see the range and skew of the data

We can provide more details in text and can include the figure below to display the data.

[Figure]

**Results**

Fig 5 and 6 are very nice, but a more variable colour scheme would help, rather than graduated shades of blue which make it difficult to distinguish close classes

We will make the suggested change.

Fig 7 – its not immediately obvious that the y-axis scales differ here. At first glance I assumed the valleys were comparable. Could you either scale the axis, or make clear note in the caption

We will modify the axes to ensure that the y axis are the same scale.

**Discussion**

If you think the order that you increase the variables influences the outcome, could you easily test this be changing the order and measuring the effect?

We have done this for one scenario, where we include increased earthquake annual frequency for Scenario 7 and Scenario 9. For example, the cumulative increase in risk associate with including a time dependent earthquake scenario in Scenario 7 is 330 % and 56 % for Franz Josef and Fox respectively, while for Scenario 9 it is 260% and 54%. In this example, we suggest that the differences are due to changes in the number of seismic landslides generated, spatial probability of impact and vulnerability. However, given that we are more concerned with the relative differences between the scenarios and therefore we think that the testing of the order of variables is out of scope and will not impact the relative differences between scenarios. We will update this point in the discussion to make this clearer.

The climate change discussion is a really interesting one and worthwhile here. However, the role of climate change on landslide rates in these areas is really complex and it's hard to confidently say what might happen – landslide frequency could drop while size increases for instance.

Agreed. We have tried to highlight this in our discussion but will expand upon this point and the associated uncertainties of increased landslide risk under climate change. In our sensitivity analysis we do not test changes in gradient of the -magnitude – frequency distributions to reflect increases in the frequency of larger landslides occurring relative to the frequency of smaller landslides. Such shifts in the magnitude – frequency distribution will impact the risk results and associated uncertainty.

One aspect missing from the discussion for me is the temporal variation in exposure. Firstly, covid may well have long term implications for visitor numbers that your values won't account for. Secondly, visitors in the valleys are no doubt much lower on rainy days when aseismic landslides are more likely than dry days, when aseismic landslides are less likely. The question is whether these changes cancel each other out. It's also not clear to me if you take diurnal variations into account or not – how many people are in the valley at night?

Will include a point regarding societal risk and the reduction in visitor numbers due to Covid. Additionally, within our societal risk calculations, detailed in Massey et al., 2018, we consider diurnal variations.

With regards to rainfall induced landsliding, we can include a theoretical reduction in aseismic landslide risk for visitor risk per trip that assumes that in heavy rainfall conditions when landslides are likely to be triggered the tracks are closed and the visitor is not present. Such a theoretical reduction was also included in the Massey et al., 2018 report. It is current DOC policy to close or partly close the tracks in each valley under heavy rainfall conditions or when heavy rain warnings are in place.

For this example (see figure below), we've set a theoretical reduction in aseismic landslide risk of 75%, assuming that 75% of our aseismic landslides are triggered under heavy rainfall conditions. However, as we cannot link our landslide inventory to rainfall events we can't provide a quantitative basis for this risk reduction number. Additionally, as mentioned earlier in our discussion section (Line 473), we may underestimate debris flow activity on the large debris fans and therefore are underestimating the rainfall induced debris flow risk on these fans. This again emphasises the need for a robust landslide inventory that captures all events. We will add in additional details and updated figure to Line 555 onwards.

[Figure]

Fig 10 – excellent, very valuable. Could you maybe add the suggested 'acceptable' risk thresholds from the ChCh rockfall work for further added context?

Happy to reference the Port Hills acceptable risk thresholds in text, but as Figure 10 plots individual risk per trip rather than AIFR we will keep this separate to avoid confusion. The $10^{-4}$ AIFR corresponds to $3 \times 10^{-7}$ risk per day

---

## Author Comment (AC3)

We thank Caroline Orchiston for the helpful suggestions and comments, as well as taking the time to read and evaluate our manuscript. Please find outlined below our response to the suggestions made on the manuscript.

L62 - Visitors are described as being able to 'easily access and experience a glacier environment '. For day visitors (unlike more dedicated trampers/mountaineers) I would suggest the access is no longer easy, and that visitors are entering what could almost be described as a post-glacial environment in the lower reaches of the valleys. In the 1990s and early 2000s it was possible for day visitors to walk to the terminus and do a Glacier Walk on a paid tour. This is no longer possible because of the extent of glacial retreat, meaning that visitors have to take a tour via helicopter to fly above the glacier and do a glacial landing in order to get a full glacier experience. Thus the risk to tourists is not just downstream of the terminus, but on the glacier itself during landings. Was this factored into your analysis? It would also be useful to cite some data on the approx. proportion of visitors (pre-Covid) doing glacier flights compared to walking up to the viewing areas in the valleys. DOC would hold this data as concession manager.

We will amend our Study Site description to be clearer about the day visitors (see below). This risk analysis was only concerned with visitors walking DOC tracks in the valley up until a glacier viewpoint.

*"The glaciers themselves can now only be accessed via helicopters, with visitors undertaking paid tours on the glacier, given the commercial sensitivities the risk from landslide hazards to visitors on commercial tours on the glacier have not been quantified in our risk analysis.*

Due to commercial nature of these activities, quantifying risk to visitors on the glacier themselves was outside of the scope of this study and therefore we do not provide information with regards to the proportion of visitors walking the tracks versus partaking in commercial tours.

L71 - the sentence about the Alpine Fault as a major earthquake source needs a reference – the Howarth et al. 2021 Nature Geoscience paper is the best most recent.

We will make the suggested change.

Figure 1 – the text size on c) and d) is too small.

We will make the suggested changes to the figure.

Line 140 – can you provide some background context of the nature of the workforce in the glacial valleys? What types of jobs do they do, and how frequently are they undertaking these jobs? (Noting that this analysis is reported elsewhere in Massey 2018). Are they working in a limited area, or do they range across the valleys, and roughly how much time are they exposed to risk per day?

In Massey et al., 2018 we calculated the AIFR for the most exposed DOC worker. This was the person who walked and undertook daily checks of the main visitor tracks in each valley. This ranged from 2.2 hours to 2.8 hours a day depending on the valley. We will add this detail into Line 140 onwards, but more information is provided in Massey et al., 2018. This report will be made publicly available.

Line 175 – before you talk about risk exposure of tourists, you need to provide visitor data for the valleys (DOC and Stats NZ are the best sources), and also provide some context on the past two years of low visitor international visitation due to the pandemic. The pandemic hit following several years of strong tourism growth in NZ. The glaciers, alongside Aoraki/Mt Cook and Piopiotahi/Milford Sound were experiencing a million visitors per annum, causing significant pressure on tourism infrastructure and other social/community pressures. Glacier Country / South Westland had a heavy reliance on international visitors pre-Covid and has been one of the hardest hit regions of NZ in terms of reduced visitation when the international borders closed. Unlike other parts of NZ that moved to a domestic market quite effectively, South Westland is less accessible and can't attract e.g. weekend visitors because of its remoteness. This sort of context has implications for exposure and vulnerability (i.e. internationals are less likely to speak English and thus may not understand risk communication information).

We will add in information regarding pre-covid visitor data for the valleys to the Study site section, see below:

*Prior to the Covid 19 pandemic and associated closure of New Zealand's border to international tourists, c.700,000 people per year walked the tracks in the Franz Josef Glacier Valley and c.400,000 people per year walked the tracks in Fox Glacier Valley. A maximum number of 6,000 people per day and 3,500 people per day walked the tracks in Franz Josef and Fox Glacier Valley respectively. Within this environment, visitors are exposed to a variety of landslide hazards"*

Within the Time Variable Risk section of the discussion, we will include a point on the impact of Covid on exposure and vulnerability to emphasise its not only changes in hazard but the elements at risk that are also important (see italicised paragraph below). We suggest that a reduction in visitor numbers will reduce societal risk, which we define using f/N pairs where f represents the frequency of the event occurring and N represents the number of fatalities.

Alongside the impact of Covid, both main visitor tracks in the valley have been closed or partly closed due to geomorphic processes. In Fox Glacier from 2019 onwards the main visitor track on the north side of the valley has been closed due to the impact of the Alpine Gardens Landslide (access is now only provided on the south side of the valley). In Franz Josef from 2020 onwards the Waiho river has shifted towards the true left side of the valley resulting in the partial closure of main visitor track. This change in visitor tracks will also have knock on effects to both individual visitor risk and societal risk.

*"Alongside changes in hazard behaviour, risk analysis should also account for dynamic changes in exposure and vulnerability. … Since 2019 and 2020 the main visitor tracks in the Fox and Franz Josef Glacier valleys respectively have been closed or partially closed due to geomorphic processes. Until access is restored in both valleys, the exact location of the tracks in each valley and the number of people walking the tracks is unknown. In Fox Glacier Valley, the expansion of the Mill's Creek fan from debris flow activity has damaged access on the true-right side of the valley, while in Franz Josef Glacier Valley, the course of the Waiho River has restricted access on the true-left side of the valley. As such visitor exposure and therefore risk to landslide hazard is reduced. Alongside this, the Covid-19 pandemic and associated closure of New Zealand's border to international tourists has resulted in a reduction in visitor numbers to both glacier valleys. This reduction in visitor numbers will impact our societal risk metric, by reducing exposure of 1 or more people to an event that might result in fatalities."*

We will make the suggested change.

Visitor exposure – walking speed isn't thoroughly explained and will be highly variable, e.g. family groups with young children are potentially more exposed due to frequent stopping, and slow pace.

We will add more detail to Section 3.6. Based on information provided by DOC, in Fox Glacier Valley, our average walker spent 1.5 hours walking to and from the glacier viewpoint and 0.2 hours driving to and from the car park (see Figure 1 c), while our slower walker spent 2 hours walking to and from the glacier viewpoint and 0.3 hours driving to and from the car park. In Franz Josef Glacier Valley, our average walker spent 2-hour walking to and from the glacier view point, and 0.3 hours driving to and from the car park (see Figure 1 d) while the slower walker spent 2.5 hours walking to and from the glacier viewpoint and 0.4 hours driving to and from the car park.

We also conducted our own field counts and checks on walking speed and approx. visitor numbers. This also revealed that not all visitors travelled the full length of the tracks and turned back at certain points. More detailed analysis and investigation of visitor behaviour and its impact on both exposure and vulnerability would be interesting to include but outside the scope of this study.

Figure 5a – the entirely blue colour gradient makes it very difficult to discern specific risk zones. Also 5c has a colour legend which is hard to match to the chart. Likewise with 6c.

We will make the suggested changes to the colour ramps and legends in both figures.

We will make the suggested change.

Use of the term societal risk – it is unclear. E.g Individual risk is clearly risk present to one person, but societal risk speaks to a much broader impact, while here you use it to describe risk to N or more people, i.e. mulitiple injuries or fatalities. Consider a different term.

We will define in Section 3.2 what we are defining as societal risk, which is the frequency of events resulting in 1 or more fatalities, as well as what we are excluding from societal risk (e.g. the broader impacts).

Throughout the document we will also make it clear when referring to societal risk that it is only the probability of multiple fatalities occurring.

Risk communication – Fig 10 is a useful way to attempt to convey comparative risks across commonly used activities. Effective communication of technical, complex risk scores with big numbers/exponents and uncertainties in order to support policy and practice is a huge challenge. While it is beyond the scope of this paper, the findings offer very interesting food for thought on how we can improve the translation of scenario-based risk data to maximise the societal benefits from a disaster risk reduction perspective. The use of qualitative measures/scales to translate risk into risk acceptability/tolerance data would be useful, but would require a detailed study of the local stakeholder and visitor perceptions of risk in order to generate appropriate risk tolerance outputs.

Thanks- this is an important point and comment, though outside the scope of this paper as noted. We have changed a sentence in Line 608 to highlight and provide a nod towards the role of risk perception (and the wider subject) in setting risk thresholds:

*"The risk bands can be presented against risk comparator data to inform risk evaluation and risk tolerability processes in conjunction with an evaluation of how visitors and decision makers perceive risk (cf. Taig, 2022)."*

The Taig 2022 reports contain more information on the risk comparator data and provide guidance to DOC for setting risk thresholds. It is our understanding that DOC intends to make these reports available on their website. The references for the reports are:

Taig T. 2022a. Risk comparisons for DOC visitors and staff. Cheshire (GB): TTAC Ltd. 131 p.

Taig T. 2022b. Guidelines for DOC on dealing with natural hazard risk. Cheshire (GB): TTAC Ltd. 91 p.